# The functional form of value normalization in human reinforcement learning

Sophie Bavard[1,2,3]*, Stefano Palminteri[1,2]*

[1]Laboratoire de Neurosciences Cognitives et Computationnelles, Institut National de la Santé et Recherche Médicale, Paris, France; [2]Département d'Etudes Cognitives, Ecole Normale Supérieure, PSL University, Paris, France; [3]Department of Psychology, University of Hamburg, Hamburg, Germany

**Abstract** Reinforcement learning research in humans and other species indicates that rewards are represented in a context-dependent manner. More specifically, reward representations seem to be normalized as a function of the value of the alternative options. The dominant view postulates that value context-dependence is achieved via a divisive normalization rule, inspired by perceptual decision-making research. However, behavioral and neural evidence points to another plausible mechanism: range normalization. Critically, previous experimental designs were ill-suited to disentangle the divisive and the range normalization accounts, which generate similar behavioral predictions in many circumstances. To address this question, we designed a new learning task where we manipulated, across learning contexts, the number of options and the value ranges. Behavioral and computational analyses falsify the divisive normalization account and rather provide support for the range normalization rule. Together, these results shed new light on the computational mechanisms underlying context-dependence in learning and decision-making.

**\*For correspondence:**
sophie.bavard@gmail.com (SB);
stefano.palminteri@ens.fr (SP)

**Competing interest:** The authors declare that no competing interests exist.

## Editor's evaluation

This important study presents a series of behavioral experiments that test whether value normalization during reinforcement learning follows divisive or range normalization. Behavioral data from probe tests with two alternatives demonstrate convincingly that range normalization provides a better account for choice behavior and value ratings in this setting. These findings will be of interest for readers interested in neuroeconomics and cognitive neuroscience.

## Introduction

The process of attributing economic values to behavioral options is highly context-dependent: the representation of an option's utility does not solely depend on its objective value, but is strongly influenced by its surrounding (i.e., other options simultaneously or recently presented). This is true in an extremely wide range of experimental paradigms, ranging from decision among lotteries to reinforcement learning problems (*Kahneman and Tversky, 1984*; *Huber et al., 1982*; *Klein et al., 2017*; *Bavard et al., 2018*; *Spektor et al., 2019*). This is also true for a wide variety of species, including mammals (*Yamada et al., 2018*; *Conen and Padoa-Schioppa, 2019*), birds (*Pompilio and Kacelnik, 2010*) and insects (*Pompilio et al., 2006*; *Solvi et al., 2022*). The pervasiveness of this effect across tasks and species suggests that context-dependence may reflect the way neuron-based decision systems address a fundamental computational trade-off between behavioral performance

and neural constraints (*Fairhall et al., 2001*; *Padoa-Schioppa, 2009*; *Kobayashi et al., 2010*; *Louie and Glimcher, 2012*).

Indeed, it has been showed that context-dependence often takes the form of a normalization process where option values are rescaled as a function of the other available options, which has the beneficial consequence of adapting the response to the distribution of the outcomes (*Louie and Glimcher, 2012*). The idea that neural codes and internal representations are structured to carry as much as information per action is the cornerstone of the efficient coding hypothesis, demonstrated both at the behavioral and neural levels, in perceptual decision-making (*Reynolds and Heeger, 2009*).

Probably due to its popularity in perception neuroscience, the dominant view regarding the computational implementation of value normalization in economic decisions postulates that it follows a divisive rule, according to which the subjective value of an option is rescaled as a function of the sum of the value of all available options (*Louie et al., 2011*; *Louie et al., 2013*; *Louie et al., 2015*; *Webb et al., 2021*; *Pirrone and Tsetsos, 2022*). In addition, to be validated in the perceptual domain, the divisive normalization rule also presents the appeal of being reminiscent of Herrnstein's matching law for behavioral allocation (*Herrnstein, 1961*).

Even though, to date, most of the empirical studies proposing divisive normalization as a valid model of economic value encoding proposed that option values are vehiculated by explicit features of the stimulus (such as food snacks or lotteries: so-called described options; *Hertwig and Erev, 2009*; *Louie et al., 2013*; *Garcia et al., 2021*; *Daviet and Webb, 2023*), few recent studies have extended the framework to account for subjective valuation in the reinforcement learning (or experience-based) context (*Juechems et al., 2022*; *Louie, 2022*). Adjusting the divisive normalization model to a reinforcement learning scenario is easily achieved by assuming that the normalization step occurs at the outcome stage, that is, when the participant is presented with the obtained (and forgone) outcomes.

While predominant in the current neuroeconomic debate about value encoding and adaptive coding (*Bucher and Brandenburger, 2022*), the divisive normalization account of value normalization is not consensual (*Padoa-Schioppa, 2009*; *Kobayashi et al., 2010*; *Padoa-Schioppa and Rustichini, 2014*; *Burke et al., 2016*; *Gluth et al., 2020*). Indeed, range normalization represents a possible alternative account of value normalization and is made plausible by both behavioral and neural observations (*Parducci, 1963*; *Rustichini et al., 2017*). According to the range normalization rule, option values are rescaled as a function of the maximum and the minimum values presented in a context, irrespective of the number of options or outcomes (or set size; *Conen and Padoa-Schioppa, 2019*; *Parducci, 1963*; *Bavard et al., 2021*). Answering this question bears important consequences for neuroscience because understanding the scaling between objective and subjective outcomes is paramount to investigate the neural codes of economic values and understand the neural mechanisms of decision-making (*Cox and Kable, 2014*; *Lebreton et al., 2019*). Yet, the experimental paradigms used so far in reinforcement learning research were ill-suited to distinguish between two accounts of value normalization in the context of reinforcement learning (*Klein et al., 2017*). To address this issue, we designed a new reinforcement learning protocol where, by simultaneously manipulating outcome ranges and choice set sizes, we made the divisive and the range normalization predictions qualitatively diverge in many respects (*Roberts and Pashler, 2000*; *Palminteri et al., 2017*.) We opted for a reinforcement learning paradigm because it has a greater potential for translational and cross-species research (*Garcia et al., 2021*). In a total of eight experiments (N = 500 in total), we deployed several variations of this new behavioral protocol where we controlled for several factors. The behavioral, model fitting and simulation results convergently rejected divisive normalization as a satisfactory explanation of the results in favor of the range normalization account. Results also suggested that the range normalization account should be further improved by a nonlinear weighting process. To check the robustness of our results across different elicitation methods and representational systems, we also assessed option values using explicit ratings. Values inferred from explicit, declarative, ratings were remarkably consistent with those inferred from more traditional, choice-based, methods.

## Results

### Computational hypotheses and ex ante model simulations

The goal of this study was to characterize the functional form of outcome (or reward) normalization in human reinforcement learning. More specifically, we aimed at arbitrating between two equally

plausible hypotheses: range normalization and divisive normalization. Both hypotheses assume that after reception of a given objective reward $R$, the learner forms an internal, subjective, representation of it, $R_{NORM}$, which is influenced by other contextually relevant rewards. Crucially, the two models differ in how $R_{NORM}$ is calculated. According to the range normalization hypothesis, the subjective normalized reward $R_{NORM}$ is defined as the position of the objective reward $R$ within its contextual range:

$$R_{NORM} = \frac{R - R_{MIN}}{R_{MAX} - R_{MIN}} \tag{1}$$

where $R_{MAX}$ and $R_{MIN}$ are the endpoints of the contextually relevant distribution and together form the range ($R_{MAX} - R_{MIN}$). On the other side, the divisive normalization hypothesis, in its simplest form, postulates that the subjective normalized reward $R_{NORM}$ is calculated by dividing the objective reward by the sum of all the other contextually rewards (*Louie, 2022*):

$$R_{NORM} = \frac{R}{\sum_{k=1}^{n} R_k} \tag{2}$$

where $n$ is the number of contextually relevant stimuli. These hypotheses concerning value normalization are then easily plugged into the reinforcement learning framework, simply by assuming that the value of an option is updated by minimizing a prediction error, calculated on the basis of the subjective reward. Although these normalization functions are mathematically distinct, they make identical (or very similar) behavioral predictions in many of the experimental protocols designed to investigate context-dependent reinforcement learning so far (*Klein et al., 2017*; *Bavard et al., 2018*; *Spektor et al., 2019*; *Bavard et al., 2021*; *Palminteri et al., 2015*). It should be noted here that, although divisive normalization has been more frequently applied to the prospective evaluation of described

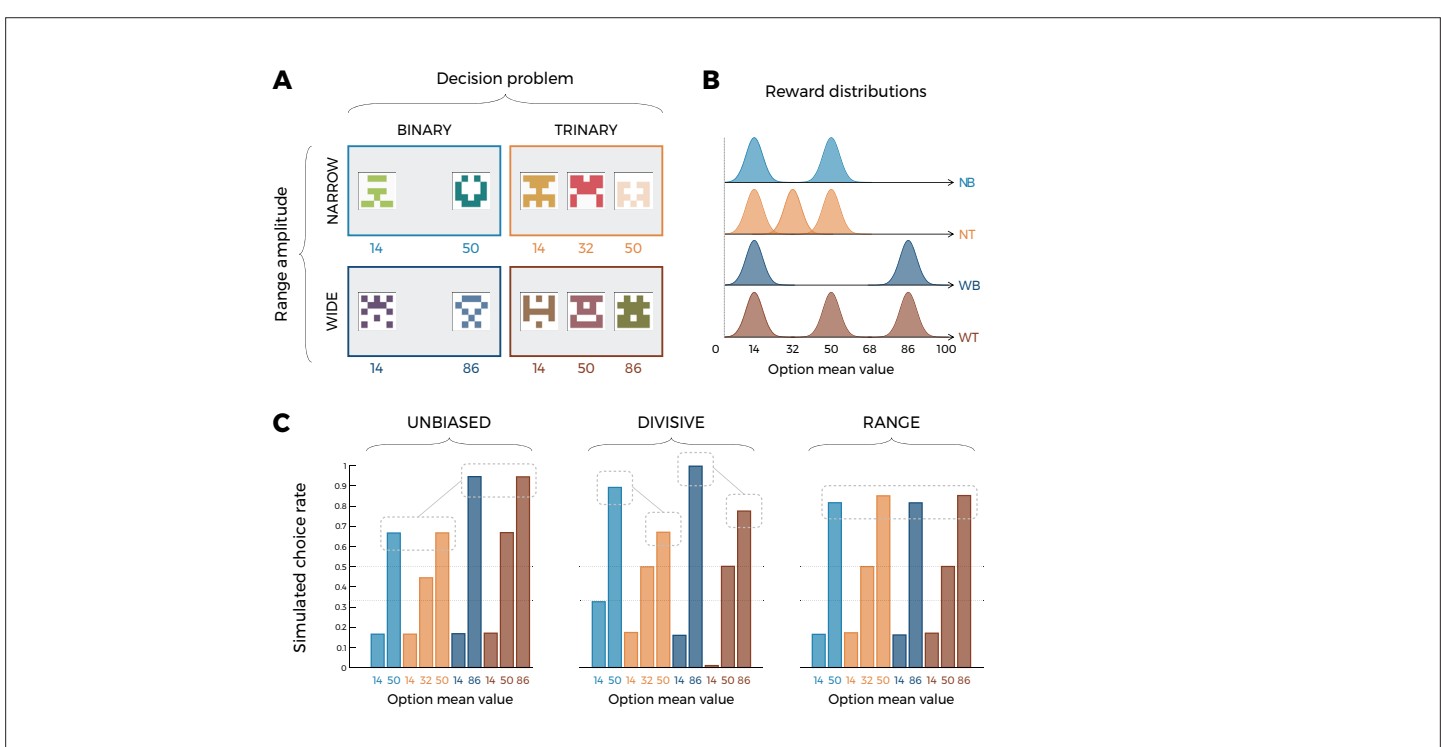

**Figure 1.** Experimental design and model predictions of Experiment 1. (**A**) Choice contexts in the learning phase. Participants were presented with four choice contexts varying in the amplitude of the outcomes' range (narrow or wide) and the number of options (binary or trinary decisions). (**B**) Means of each reward distribution. After each decision, the outcome of each option was displayed on the screen. Each outcome was drawn from a normal distribution with variance $\sigma^2 = 4$. NB: narrow binary, NT: narrow trinary, WB: wide binary, WT: wide trinary. (**C**) Model predictions of the transfer phase choice rates for the UNBIASED (left), DIVISIVE (middle), and RANGE (right) models. Note that choice rate in the transfer phase is calculated across all possible binary combinations involving a given option. While score is proportional to the agent's preference for a given option, it does not sum to one because any given choice counts for the final score of two options. Dashed lines represent the key prediction for each model.

outcomes (e.g., lotteries; snack-food items), rather than retrospective evaluation of obtained outcomes (e.g., bandits), it has both historical (*Herrnstein, 1961*) and recent (*Louie, 2022*) antecedents in the context of reinforcement learning. For the present study, we designed reinforcement learning tasks designed to adjudicate these computational hypotheses. The key idea behind our behavioral protocol is to orthogonally manipulate, across different learning contexts, the amplitude of the range of the possible outcomes and the number of options (often referred to as choice size set; *Figure 1A*; see also *Figure 2—figure supplement 2A* for an alternative version). The first factor (the amplitude of the range of the possible outcomes) is key to differentiate an unbiased model (where $R_{NORM} = R$) from both our normalization models. The second factor (number of options) is key to differentiate the range normalization from divisive normalization. The reason for this can easily be inferred from *Equations 1 and 2* because adding more outcomes has a significant impact on the subjective reward $R_{NORM}$ only following the divisive normalization rule (*Daviet and Webb, 2023*; as clearly put by the main advocate of the divisive normalization rule for value-based decision-making: "[…] a system would be highly sensitive to the number of options under consideration. As the number of elements in the denominator grows, so does the aggregate value of the denominator, shifting the overall firing rates lower and lower" (*Glimcher, 2022*, page 14). To quantitatively substantiate these predictions, we ran model simulations using three models. We compared a standard model with unbiased subjective values (UNBIASED), and two normalization models using either the divisive or the range normalization rules (referred to as DIVISIVE and RANGE, respectively). First, we simulated a learning phase, where each learning context in our factorial design was presented 45 times. After each trial, the simulated agent was informed about the outcomes that were drawn from normal distributions (*Figure 1B*). To avoid sampling issues and ambiguity concerning the definition of the relevant normalization variables, the simulated agents were provided information about the outcomes of all options ('complete' feedback; *Hertwig and Erev, 2009*; *Li and Daw, 2011*). After the *learning* phase, the simulated agents went through a *transfer* phase, where they made decisions among all possible binary combinations of the options (without additional feedback being provided). Similarly constructed experiments, coupling a learning to a transfer phase, have been proven key to demonstrate contextual effects in previous studies (*Klein et al., 2017*; *Bavard et al., 2018*; *Pompilio and Kacelnik, 2010*; *Bavard et al., 2021*; *Palminteri et al., 2015*; *Hayes and Wedell, 2022*; *Juechems et al., 2022*). When analyzing model simulations, we focused on choice patterns in the transfer phase (of note, accuracy during the learning phase is weakly diagnostic because all models predict above chance accuracy and, to some extent, a choice size set effect, whose level depends on the choice stochasticity parameter of the softmax decision rule). *Figure 1C* plots the average simulated choice rate in the transfer phase. For a given option, the transfer phase choice rate was calculated by dividing the number of times an option is chosen by the number of times the option is presented. In the transfer phase, the 10 cues from the learning phase were presented in all possible binary combinations (45, not including pairs formed by the same cue). Each pair of cues was presented four times, leading to a total of 180 trials. Since a given comparison counts for the calculation of the transfer phase choice rate of both involved options, this implies that this variable will not sum to one. Nonetheless, the relative ranking between transfer choice rate can be taken as a behavioral proxy of their subjective values.

Crucially, even if the transfer phase involves only binary choices, it can still tease apart the normalization rules affecting outcome valuation during the learning phase. This is because transfer choices are made based on the memory of values acquired during the learning phase, where we purposely manipulated the number of options and their ranges of values, in order to create learning contexts that allow to confidently discriminate between the two normalization accounts, in the reinforcement learning context.

Unsurprisingly, within each learning context, in all models the choice rates are higher for high-value options compared to lower value options. However, model simulations show that the models produce choice patterns that differ in many key aspects. Let's start considering the UNBIASED model as a benchmark (*Figure 1C*, left). Since it encodes outcomes in an unbiased manner, it predicts higher choice rates for the high-value option in the 'wide' contexts (WB$_{86}$ and WT$_{86}$) compared to high-value options in the 'narrow' contexts (NB$_{50}$ and NT$_{50}$). On the other side, the UNBIASED model predicts that choice rate in the transfer phase is not affected by whether or not the option belonged to a binary or a trinary learning context. Moving to the DIVISIVE model, we note that the difference between the choice rates of high-value options of the 'wide' contexts (WB$_{86}$ and WT$_{86}$) compared

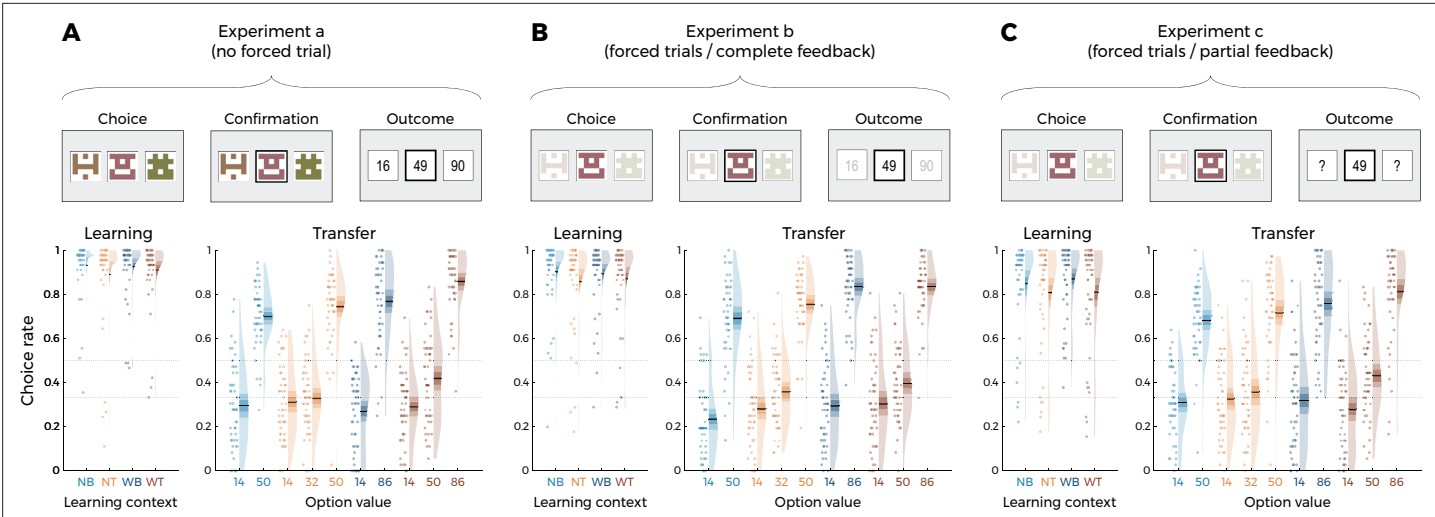

**Figure 2.** Behavioral results of Experiment 1. Top: successive screens of a typical trial for the three versions of the main experiment: without forced trials (**A**), with forced trials and complete feedback information (**B**) and with forced trials and partial feedback information (**C**). Bottom: correct choice rate in the learning phase as a function of the choice context (left panels), and choice rate per option in the transfer phase (right panels) for the three versions of the main experiment: without forced trials (**A**), with forced trials and complete feedback information (**B**) and with forced trials and partial feedback information (**C**). In all panels, points indicate individual average, shaded areas indicate probability density function, 95% confidence interval, and SEM (n=50). NB: narrow binary, NT: narrow trinary, WB: wide binary, WT: wide trinary.

The online version of this article includes the following figure supplement(s) for figure 2:

**Figure supplement 1.** Behavioral results of the pilot Experiment similar to Experiments 1 and 2.

**Figure supplement 2.** Experimental design, model predictions, and behavioral results concerning Experiment 2.

those of the 'narrow' contexts ($NB_{50}$ and $NT_{50}$) is now much smaller due to the normalization process (*Figure 1C*, middle). However, the DIVISIVE model also predicts that the choice rate is strongly affected by whether or not the option belonged to a binary or a trinary learning context. For instance, $WB_{86}$ and $NB_{50}$ present a much higher choice rate compared to $WT_{86}$ and $NT_{50}$, respectively, despite their objective expected value being the same. This is an easily identifiable and direct consequence of the denominator of the divisive formulation rule increasing as a function of the number of options (*Equation 2*). Concerning the RANGE model (*Figure 1C*, right), it predicts choice rates being similar across all high-value options, regardless of their objective values (because of the normalization) and whether or not the option belonged to a trinary or binary context (because of the range normalization rule; *Equation 1*). Finally, the choice rates of the low-value (14) options also discriminate the DIVISIVE model, where it is strongly modulated by the task factors, from the other two models, where all low-value options present the same choice rate. To conclude, model simulations confirm that our design, involving a factorial learning phase and a transfer phase, is well suited to disentangle our three a priori models because they predict qualitatively differentiable patterns of choices (see also *Figure 2—figure supplement 2C* for similar conclusions based on an alternative task design; *Palminteri et al., 2017*; *Teodorescu and Usher, 2013*).

## Behavioral results

The above-described behavioral protocol was administered to N = 50 participants recruited online, who played for real monetary incentives as previously described (*Bavard et al., 2021*). We first tested whether the correct choice rate (i.e., the probability of choosing the option with the highest expected value) was overall above chance level during the learning phase to ensure that the participants engaged in the task. Indeed, correct response rate was significantly higher than chance level (0.5 and 0.33 in the binary and trinary learning contexts, respectively) in all conditions (least significant comparison: $t(49) = 18.93$, $p<0.0001$, $d = 2.68$; on average: $t(49) = 24.01$, $p<0.0001$, $d = 3.96$; *Figure 2A*). We further checked whether the task factors affected performance in the learning phase and found a significant effect of the decision problem (the correct choice rate being higher in the binary compared

to the trinary contexts: $F(1,49) = 9.26$, p=0.0038, $\eta^2 = 0.16$), but no effect of range amplitude (wide versus narrow; $F(1,49) = 0.52$, p=0.48, $\eta^2 = 0.01$) nor interaction ($F(1,49) = 2.23$, p=0.14, $\eta^2 = 0.04$).

We next turned to the results of the transfer phase. Following the analytical strategy used in previous studies, we first checked that the correct choice rate in the transfer was significantly higher than chance ($t(49) = 9.10$, p<0.0001, $d = 1.29$), thus providing positive evidence of value retrieval and generalization (*Bavard et al., 2018*; *Bavard et al., 2021*; *Hayes and Wedell, 2022*). We analyzed the choice rate per symbol, which is the average frequency with which a given symbol is chosen in the transfer phase, and can therefore be taken as a measure of the subjective preference for a given option (*Bavard et al., 2018*; *Palminteri et al., 2015*). We focus on key comparisons that crucially discriminate between competing models of normalization. First, and contrary to what was predicted by the DIVISIVE model, the choice rate for the high-value options in the trinary contexts ($NT_{50}$ and $WT_{86}$) was not lower compared to that of the binary ones ($NB_{50}$ and $WB_{86}$). Indeed, if anything, their choice rate was higher ($NT_{50}$ vs. $NB_{50}$: $t(49) = 1.66$, p=0.10, $d = 0.29$; $WT_{86}$ vs. $WB_{86}$: $t(49) = 2.80$, p=0.0072, $d = 0.53$). Similarly, the choice rate of the low-value options was not affected by their belonging to a binary or trinary context in the direction predicted by the DIVISIVE model. Concerning other features of the transfer phase performance, some comparisons were consistent with the UNBIASED model and not with the RANGE model, such as the fact that high-value options in the narrow contexts ($NB_{50}$ and $NT_{50}$) displayed a lower choice rate compared to the high-value options of the wide contexts ($WB_{86}$ and $WT_{86}$; $t(49) = -4.19$, p=0.00011, $d = -0.72$), even if the size of the difference appeared to be much smaller to that expected from ex ante model simulations (*Figure 1C*, right). Other features were clearly more consistent with the RANGE model. For instance, the fact that the mid-value option in the wide trinary context $WT_{50}$ displayed a significantly lower choice rate compared to the high-value options in the narrow contexts ($NT_{50}$ and $NB_{50}$) was not predicted by the UNBIASED model. One feature was not explained by any of the models, such as the higher choice rate for the high-value options in the trinary contexts ($NT_{50}$ and $WT_{86}$) compared to the binary contexts ($NB_{50}$ and $WB_{86}$; $t(49) = 3.53$, p=0.00090, $d = 0.50$; please note that the statistical test stays significant when taking into account all experiments: $t(149) = 4.11$, p<0.0001, $d = 0.34$). Of note, the direction of the effect for this comparison is in stronger contrast with the DIVISIVE (which predicts a difference in the opposite direction) compared the RANGE and UNBIASED models (which predict no difference).

Finally, the mid-value options ($NT_{32}$ and $WT_{50}$) displayed a choice rate very close to that of the corresponding low-value options ($NT_{14}$ and $WT_{14}$): this feature is clearly in contrast with both the DIVISIVE and UNBIASED models (which predict their choice rate closer to that of the corresponding high-value options: $NT_{50}$ and $WT_{86}$), but not perfectly captured either by the RANGE model (which

**Table 1.** Experimental design.

Each version of each experiment was composed of four different learning contexts. Results of Experiments 1 and 3 are presented in the main text; results of Experiment 2 are presented in *Figure 2—figure supplement 2*. Entries inside square brackets represent the mean outcomes for the lowest, mid (when applicable), and highest value option in a given context. Concerning 'forced choices,' 'unary' refers to situations where only one option is available and the participants cannot make a choice; 'binary' refers to situations where the participant can choose between two out of three options (the high-value option cannot be chosen).

| | N | Learning contexts | | | | | | N forced choices(type / feedback) |
| --- | --- | --- | --- | --- | --- | --- | --- | --- |
| | | [14,50] | [14,32,50] | [50,86] | [50,68,86] | [14,86] | [14,50,86] | |
| Experiment 1a | 50 | X | X | | | X | X | 0 |
| Experiment 1b | 50 | X | X | | | X | X | 50 (unary / complete) |
| Experiment 1 | 50 | X | X | | | X | X | 50 (unary / partial) |
| Experiment 2a | 50 | | | X | X | X | X | 0 |
| Experiment 2b | 50 | | | X | X | X | X | 50 (unary / complete) |
| Experiment 2c | 50 | | | X | X | X | X | 50 (unary / partial) |
| Experiment 3a | 100 | | X | | | | X | 90 (binary / complete) |
| Experiment 3b | 100 | | X | | | | X | 135 (binary / complete) |

**Table 2.** Quantitative model comparison in Experiments 1 and 2.
Values reported here represent mean ± SD and median of out-of-sample log-likelihood for each model.

| Model | Experiment 1 (N = 150) Out-of-sample log-likelihood | | Experiment 2 (N = 150) Out-of-sample log-likelihood | |
| --- | --- | --- | --- | --- |
| | Mean ± SD | Median | Mean ± SD | Median |
| UNBIASED | −275.31 ± 268.75 | −162.53 | −227.24 ± 269.72 | −125.40 |
| DIVISIVE | −143.38 ± 70.40 | −124.91 | −159.89 ± 65.20 | −141.07 |
| RANGE | −116.72 ± 57.91 | −109.23 | −109.71 ± 43.91 | −106.83 |
| RANGE ($\omega$) | −97.70 ± 55.52 | −78.73 | −91.99 ± 37.79 | −79.57 |

predicts their choice rate exactly in between those of high- and low-value options). To rule out that this effect was not due to a lack of attention for the low- and mid-value options, we designed two additional experiments where we added forced-choice trials to focus the participants' attention on all possible options (*Table 1*; *Chambon et al., 2020*). In one experiment (N = 50), forced-choice trials were followed by complete feedback (*Figure 2B*), in another experiment (N = 50) forced-choice trials were followed by partial feedback (*Figure 2C*). Focusing participants' attention to all possible outcomes by forcing their choice did not significantly affect the behavioral performance neither in the learning phase (*F*(2,147) = 2.75, p=0.067, $\eta^2$ = 0.04, Levene's test *F*(2,147) = 2.43, p=0.092) nor in the transfer phase (*F*(2,147) = 0.64, p=0.53, $\eta^2$ = 0.00, Levene's test *F*(2,147) = 0.64, p=0.53). This suggests that the choice rates of the mid options reflect their underlying valuation (rather than lack of information). Given the absence of detectable differences across experiments, in the model-based analyses that follow, we pooled the three experiments together. To sum up, behavioral results, specifically in the transfer phase, are in contrast with the predictions of the DIVISIVE model and are rather consistent with the range normalization process proposed by the RANGE model. Behavioral results in three experiments (N = 50 each) featuring a slightly different design, where we added a mid-value option (NT$_{68}$) between NT$_{50}$ and NT$_{87,}$ converge to the same broad conclusion: the behavioral pattern in the transfer phase is largely incompatible with that predicted by outcome divisive normalization during the learning phase (*Figure 2—figure supplement 2*). In the following section, we substantiate these claims by formal model comparison and ex post model simulations analysis.

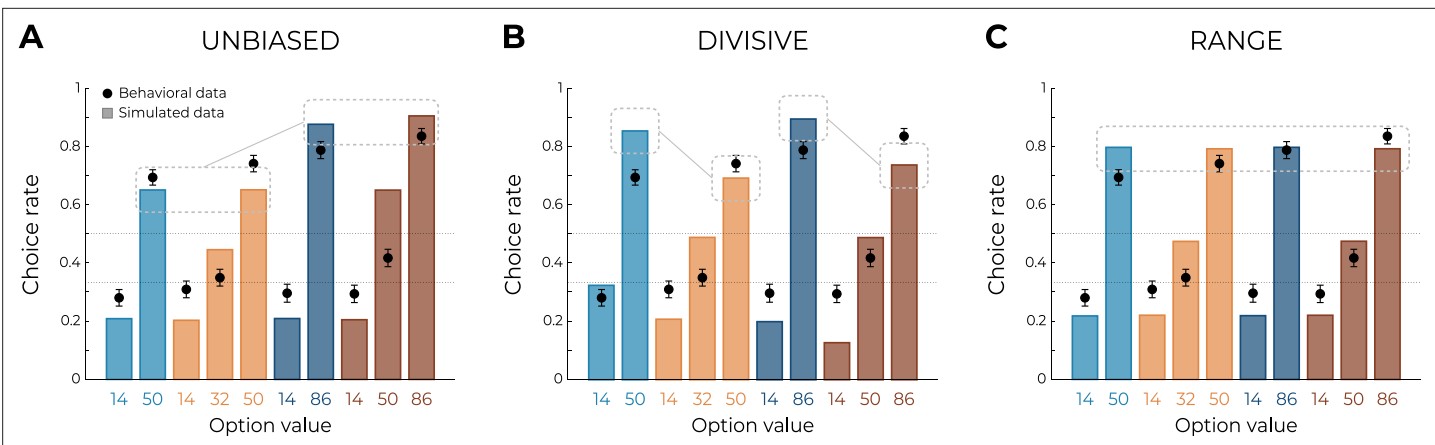

**Figure 3.** Qualitative model comparison. Behavioral data (black dots, n=150) superimposed on simulated data (colored bars) for the UNBIASED (**A**), DIVISIVE (**B**), and RANGE (**C**) models. Simulated data in the transfer phase were obtained with the best-fitting parameters, optimized on all four contexts of the learning phase. Dashed lines represent the key prediction for each model.

The online version of this article includes the following figure supplement(s) for figure 3:

**Figure supplement 1.** Ruling out more complex forms of divisive normalization.

**Figure supplement 2.** Choice rates per option in the transfer phase and model simulations.

**Figure supplement 3.** Model attributions across participants.

## Model comparison and ex post model simulations

Behavioral analyses of transfer phase choices suggest that learning and valuation are more consistent with the predictions of the RANGE model compared to those of the UNBIASED or the DIVISIVE model. To quantitatively substantiate this claim, we formally compared the quality of fit of the three models using an out-of-sample log-likelihood (*Wilson and Collins, 2019*). Specifically, we first optimized the models' free parameters (learning rates and choice inverse temperature) in order to maximize the log-likelihood of observing the learning phase choices, given the model and the parameters. We then used these parameters to generate the log-likelihood of observing the choices in the transfer phase, which were not included in the original model fitting. The RANGE model displayed a much higher mean and median out-of-sample log-likelihood (which indicated better fit) compared to both the DIVISIVE and the UNBIASED models (oosLL$_{RAN}$ vs. oosLL$_{DIV}$: $t(149) = 10.10$, p<0.0001, $d = 0.41$; oosLL$_{RAN}$ vs. oosLL$_{UNB}$: $t(149) = 8.34$, p<0.0001, $d = 0.82$; *Table 2*). Subsequently, we simulated transfer choice phase using the best fitting, that is, empirical, parameter values (*Figure 3*). The results of this ex post simulations confirmed what was inferred from the ex ante simulations and indicated that the RANGE model predicted results much closer to the observed ones, in respect of many key comparisons. All these results were replicated in three additional experiments feature with slightly different design, where the DIVISIVE model displayed a higher mean log-likelihood compared to the UNBIASED model, indicating no robust improvement in the quality of fit (see *Table 2*). Despite the superiority of the RANGE model in terms of both predictive (out-of-sample log-likelihood) and generative (simulation) performance (*Wilson and Collins, 2019*) compared to the UNBIASED and DIVISIVE one, it still failed to perfectly capture transfer phase preference, specifically concerning the mid-value options. In the subsequent section, we propose how the RANGE model could be further improved to obviate this issue.

## Improving the RANGE model

Although model comparison and model simulation both unambiguously favored the RANGE model over the UNBIASED and DIVISIVE models, the RANGE model is not perfect at predicting participants'

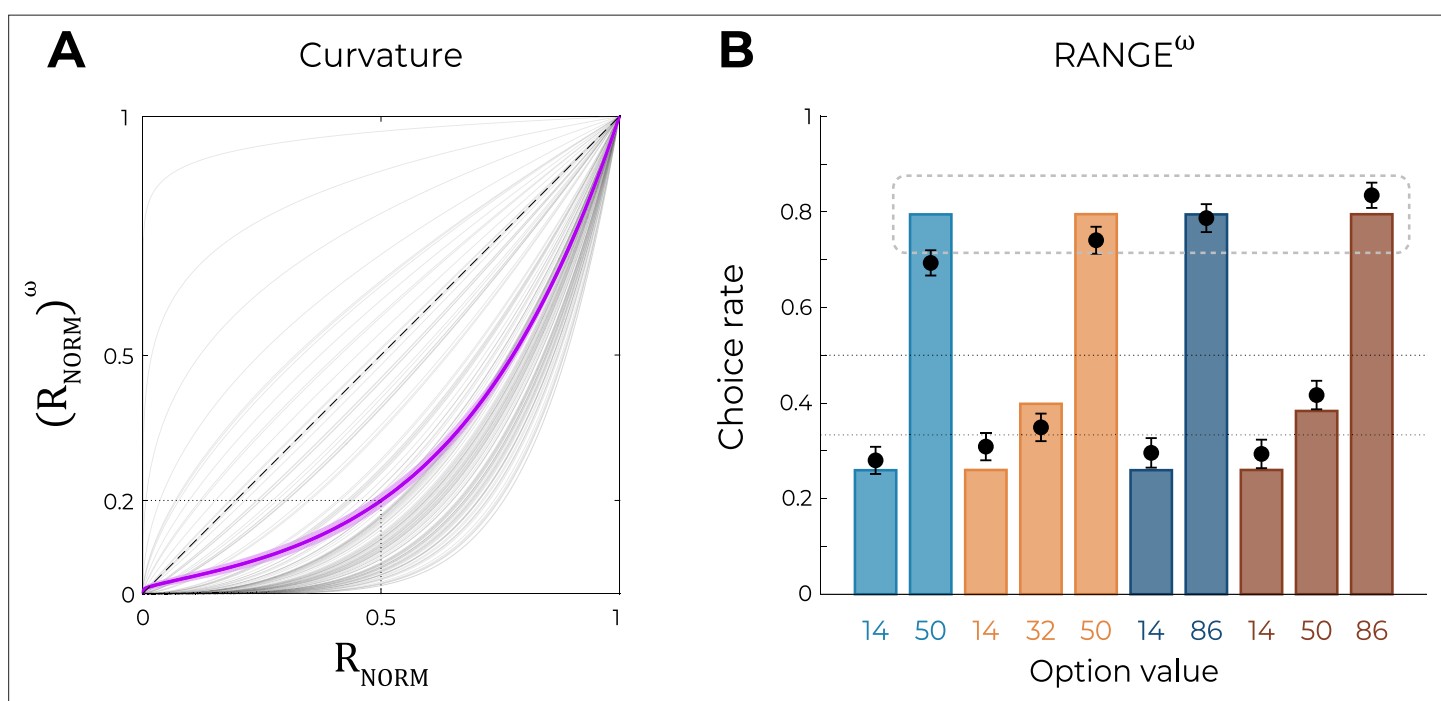

**Figure 4.** Predictions of the nonlinear RANGE model. (**A**) Curvature function of the normalized reward per participant. Each gray line was simulated with the best-fitting power parameter $\omega$ for each participant. Dashed line represents the identity function ($\omega = 1$), purple line represents the average curvature over participant, and shaded area represents SEM. (**B**) Behavioral data (black dots, n=150) superimposed on simulated data (colored bars) for the RANGE$^{\omega}$ model. Simulated data in the transfer phase were obtained with the best-fitting parameters, optimized on all four contexts of the learning phase. Dashed lines represent the key prediction for the model.

choices in the transfer phase (*Figure 3C*). As mentioned previously, the mid-value options in trinary contexts ($NT_{32}$ and $WT_{50}$) displayed a choice rate closer to that of the corresponding low-value options ($NT_{14}$ and $WT_{14}$): a feature that was not captured by the RANGE model, which predicts their choice rate to be exactly halfway of those of low-value ($NT_{14}$ and $WT_{14}$) and high-value ($NT_{50}$ and $WT_{86}$) options. In addition, the choice rate of low-value options of all contexts ($NB_{14}$, $NT_{14}$, $WB_{14}$, and $WT_{14}$) was underestimated by the RANGE model. These observations are *prima facie* compatible with the idea that outcomes are not processed linearly (*Bernoulli, 1738*; *Ludvig et al., 2018*). To formally test this hypothesis with the goal of improving the RANGE model, we augmented it with a free parameter $\omega$ that applies a nonlinear transformation to the normalized outcome. More specifically, in this modified RANGE model (RANGE$^\omega$), the normalized outcome is power-transformed by the $\omega$ parameter ($0 < \omega < +\infty$) as follows:

$$R_{NORM} = \left( \frac{R - R_{MIN}}{R_{MAX} - R_{MIN}} \right)^{\omega} \tag{3}$$

Crucially, for $\omega = 1$, the RANGE$^\omega$ reduced to the RANGE model; for $\omega < 1$, the RANGE$^\omega$ model induces a concave deformation of the normalized outcome; for $\omega > 1$, the RANGE$^\omega$ model induces a convex deformation of the normalized outcome. Quantitative model comparison favored the RANGE$^\omega$ model over all other models, including the RANGE model (*Table 2*) (oosLL$_{RAN}$ vs. oosLL$_{RAN(\omega)}$: $t(149) = -6.98$, p<0.0001, $d = -0.57$; *Table 2*). The inspection of model simulations confirmed that the RANGE$^\omega$ model closely captures participants' behavior in the transfer phase. More specifically, the mid-value options ($NT_{32}$ and $WT_{50}$) and the low-value options ($NB_{14}$, $NT_{14}$, $WB_{14}$, and $WT_{14}$) were better estimated in all contexts (*Figure 4A*; this was also true for Experiment 2; see *Figure 6—figure supplement 1*). On average, the power parameter $\omega$ was >1 (mean ± SD: 2.97 ± 1.36, $t(149) = 17.81$, p<0.0001, $d = 1.45$), suggesting that participants value the mid outcome less than the midpoint between the lowest and highest outcomes (i.e., closer to the low-value option, *Figure 4B*).

## Investigating the attentional mechanisms underlying weighted normalization

However, our current design does not allow to tease apart two possible mechanisms underlying subjective weighting of outcome captured by power transformation. One possibility (implicit in the formulation we used) is that participants 'perceive' mid outcomes as being closer to the low one because the high outcome 'stands out' due to its value. Another possibility is that participants give a higher subjective weighting to chosen outcomes because of the very fact that they were chosen and obtained. The current design and results do not allow to tease apart these interpretations because during the learning phase the mid-value options were chosen as much as the low-value options (7.2% and 6.8%, $t(149) = 0.97$, p=0.33, $d = 0.04$) and therefore mid outcomes were almost systematically unchosen outcomes.

To address this issue, we ran two additional experiments (Experiments 3a and 3b), featuring, as before, wide and narrow learning contexts (*Figure 5A*). The key manipulation in this new experiment consisted of learning contexts where we interleaved trinary choices with binary choices, where the high-value option was presented but not available to the participant (*Figure 5B*). We reasoned that by doing so we would be able to increase the number of times the mid-value options were chosen. The manipulation was successful in doing so: in the learning contexts featuring binary choices, the mid-value options were chosen on 48% of the trials (Experiment 3a) and 67% (Experiment 3b); significantly more than the corresponding high-value option in the same learning context (Experiment 3a, wide: $t(99) = 6.03$, p<0.0001, $d = 0.95$; narrow $t(99) = 5.43$, p<0.0001, $d = 0.80$; Experiment 3b, wide: $t(99) = 33.27$, p<0.0001, $d = 4.47$; narrow $t(99) = 34.06$, p<0.0001, $d = 4.33$; *Figure 5—figure supplement 1*).

We then turned to the analysis of the transfer choices and found that the manipulation was also effective in manipulating the mid-value option, so that in the contexts featuring binary choices (i.e., impossibility of choosing the high-value options), the mid options were valued more compared to the full trinary contexts (i.e., when they were almost never chosen) (Experiment 3a, wide: $t(99) = 22.80$, p<0.0001, $d = 3.46$; narrow: $t(99) = 20.10$, $d = 3.06$, p<0.0001; Experiment 3b, wide: $t(99) = 21.96$, p<0.0001, $d = 3.88$; narrow $t(99) = 20.46$, p<0.0001, $d = 3.76$; *Figure 5C*). Interestingly, the results were virtually identical in the experiment with 50% and that with 25% trinary trials despite the

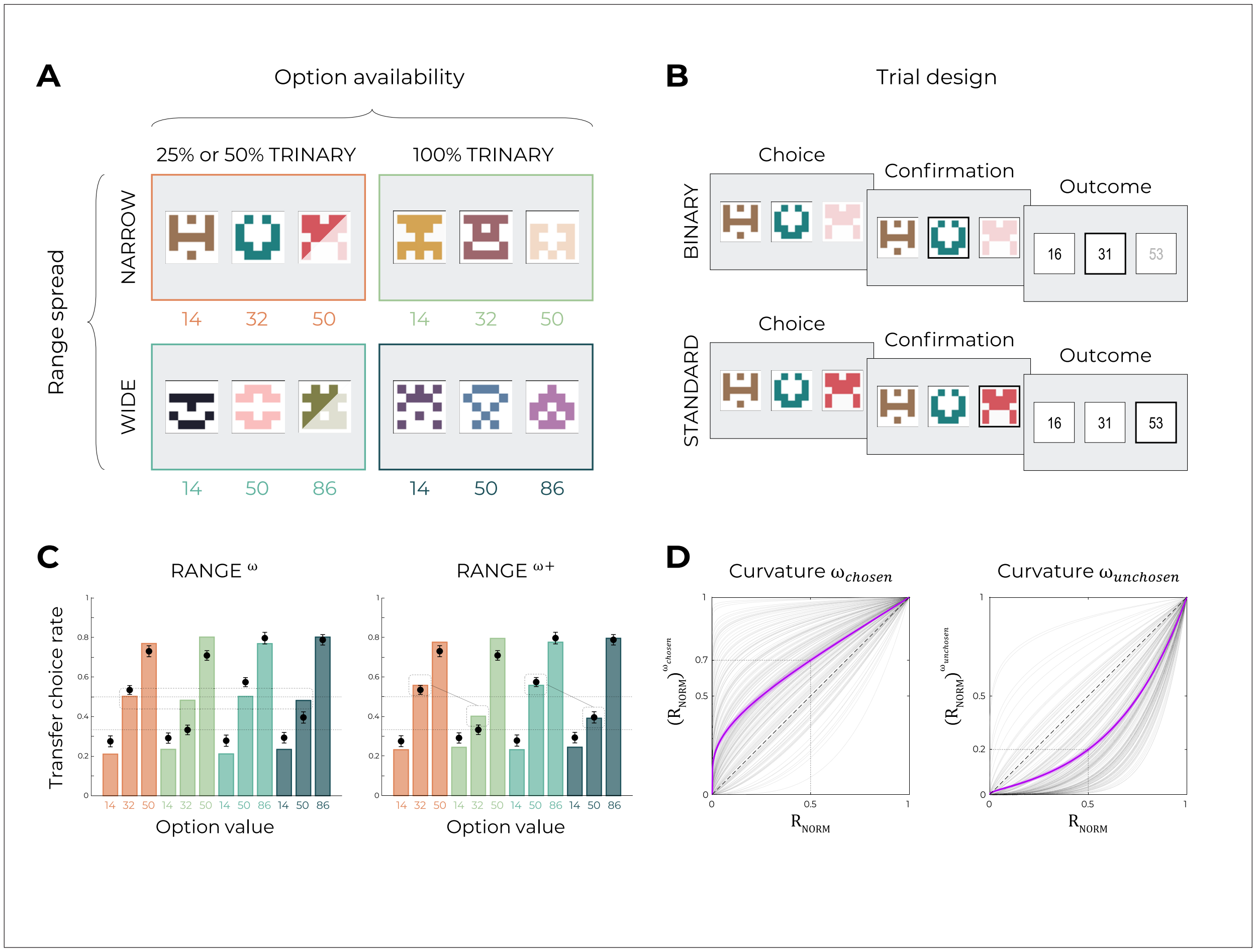


**Figure 5.** Experimental design and main results of Experiment 3. (**A**) Choice contexts in the learning phase. Participants were presented with four choice contexts varying in the amplitude of the outcomes' range (narrow or wide) and the number of available options (trinary or binary decisions). (**B**) Trial sequence for a binary trial (50 or 75% of the total number of learning trials), where the high-value option was presented but not available to the participant, and a standard trinary trial (50 or 25% of the total number of learning trials). (**C**) Behavioral data (black dots, n=200) superimposed on simulated data (colored bars) for the RANGE$^{\omega}$ and RANGE$^{\omega+}$ models. Simulated data in the transfer phase were obtained with the best-fitting parameters, optimized on all four contexts of the learning phase. Dashed lines represent the key prediction for the model. (**D**) Curvature functions of the normalized reward per participant. Each gray line was simulated with the best-fitting power parameters $\omega_c$ and $\omega_u$ for each participant. Dashed line represents the identity function ($\omega = 1$), purple line represents the average curvature over participant, and shaded area represents SEM. Results in (**C**) and (**D**) are pooled data for Experiments 3a and 3b.

The online version of this article includes the following figure supplement(s) for figure 5:

**Figure supplement 1.** Design and behavioral results of Experiment 3.

choosiness of the high-value options being very different in the two experiments. In addition, the signatures of range adaptation (narrow vs. wide) being replicated, we pooled the experiments in the main figure.

The behavioral results thus suggest that mid outcomes, although range normalized, *can* be valued correctly in between the lowest and the highest outcome if we force choices toward the mid-value option. These results are therefore consistent with the hypothesis that outcome weighting is contingent with option choosiness rather than a bias in outcome evaluation per se. To objectify this conclusion, we compared the RANGE$^{\omega}$ previously described, with a more complex one (RANGE$^{\omega+}$) where two different power $\omega$ parameters apply to the obtained (chosen: $\omega_c$) and forgone (unchosen: $\omega_u$) outcomes. This augmented model displayed better higher quality of fit in both experiments (as proxied by the out-of-sample log-likelihood of the transfer phase; $oosLL_{RAN(\omega)}$ vs. $oosLL_{RAN(\omega+)}$: $t(199) = -7.73$, p<0.0001, $d = -0.30$). This quantitative result was backed up by model simulations analysis showing that only the RANGE$^{\omega+}$ was able to capture the change in valuation in the mid-value options

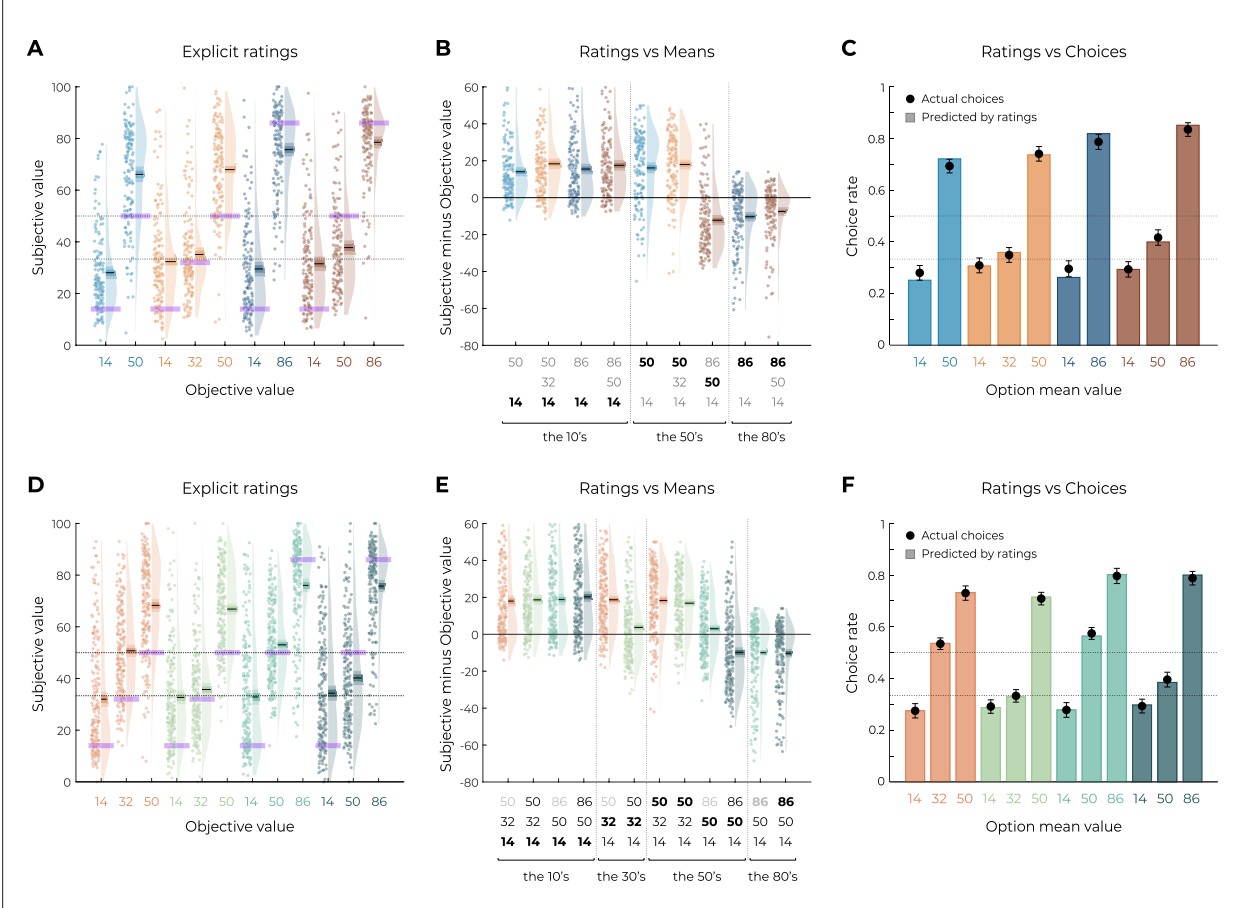

**Figure 6.** Results from the explicit elicitation phase of Experiments 1 and 3. (**A, D**) Reported subjective values in the elicitation phase for each option, in Experiment 1 (**A**, n=150) and Experiment 3 (**D**, n=200). Points indicate individual average, and shaded areas indicate probability density function, 95% confidence interval, and SEM. Purple areas indicate the actual objective value for each option. (**B, E**) Difference between reported subjective value and actual objective value for each option, arranged in ascending order, in Experiment 1 (**B**) and Experiment 3. The legend of the x-axis represents the values of the options of the context in which each option was learned (actual option value shown in bold). Points indicate individual average, and shaded areas indicate probability density function, 95% confidence interval, and SEM. (**C, F**) Behavioral choice-based data (black dots) superimposed on simulated choice-based data (colored bars), in Experiment 1 (**C**) and Experiment 3 (**F**). Simulated data were obtained with an argmax rule assuming that participants were making transfer phase decision based on the explicit subjective ratings of each option.

The online version of this article includes the following figure supplement(s) for figure 6:

**Figure supplement 1.** Qualitative model comparison and explicit ratings of Experiment 2.

---

(*Figure 5C*). Finally, we compared the weighting parameters and found $\omega_c$ significantly lower than $\omega_u$ ($t(199) = -17.28$, p<0.0001, $d = -1.92$; *Figure 5D*). To conclude, these additional experiments further clarify the cognitive mechanisms (and specifically the role of attention) underlying outcome encoding.

## Explicit assessment of option values

In addition to the transfer phase, participants performed another value elicitation assessment, where they were asked to explicitly rate the average value of each option using a slider ranging from 0 to 100. This explicit elicitation phase allowed us to have a complementary estimation of participants' subjective valuations of each option to compare them with the choice-based transfer phase. The subjective values elicited through explicit ratings were consistent with those elicited in the transfer phase hrough binary choices in many key aspects (*Figure 6A*). Indeed, against what was predicted by the DIVISIVE principle, option subjective values did not depend on the number of options in each context, but rather on their ordinal value within the context (minimum, mid, maximum). This pattern is even clearer when looking at the difference between reported subjective values and the objective values of each option (*Figure 6B*). Crucially, the subjective value of the options with an objective

value of 50 (NB$_{50}$, NT$_{50}$, and WT$_{50}$) is completely determined by its position in the range of its context, and not by the total sum of the options in this context. Finally, to compare elicitation methods, we simulated transfer phase choices based on the explicit elicitation ratings. More specifically, for each participant and comparison, we generated choices using an argmax selection rule on the subjective values they explicitly reported for each option (see *Equation 10*). We found the pattern simulated using explicit ratings to closely match the actual choice rates of the transfer phase (simulated data vs. behavioral data per option: Spearman's $\rho(8)$ = 0.99, p<0.0001, *Figure 6C*), suggesting that both elicitation methods tap into the same valuation system. Similar results and conclusions could be drawn from Experiment 2 (Spearman's $\rho(8)$ = 0.99, p<0.0001, *Figure 2—figure supplement 2*) and Experiment 3, where we confirmed that the explicit ratings of a given option were highly dependent on its position within the range (see *Figure 6D and E*). Furthermore, we also confirmed that the pattern simulated using explicit ratings closely matched the actual choice rates of Experiment 3 transfer phase (simulated data vs. behavioral data per option: Spearman's $\rho(10)$ = 1.00, p<0.0001, *Figure 6C*).

## Discussion

In this article, we sought to challenge the current dominant view of how value-related signals are encoded for economic decision-making. More precisely we designed behavioral paradigms perfectly tailored to discriminate between unbiased (or 'absolute') representations from context-dependent or normalized representations following different, antagonist, views. To do so, we deployed a series of six behavioral reinforcement learning experiments consisting of an initial learning phase (where participants learned to associate options to rewards) and a transfer – or generalization – phase allowing us to infer the subjective learned value of each option (*Palminteri and Lebreton, 2021*).

By systematically manipulating outcome ranges, we were able to confirm that behavioral data was inconsistent with the idea that humans learn and represent values in an unbiased manner. Indeed, subjective values were similar for the options presented in the decision contexts with narrow or wide decision ranges despite their objective values being very different. This result was quantitatively backed up by both model simulations and out-of-sample likelihood analyses that suggested the unbiased model being worst on average than any other normalization model. Thus, our findings significantly add up to the now overwhelming body of evidence indicating that value representations are highly context-dependent even in the reinforcement learning scenario, both in human (*Klein et al., 2017*; *Bavard et al., 2018*; *Spektor et al., 2019*; *Juechems et al., 2022*; *Bavard et al., 2021*; *Palminteri et al., 2015*; *Hayes and Wedell, 2022*) and nonhuman animals (*Yamada et al., 2018*; *Conen and Padoa-Schioppa, 2019*; *Pompilio and Kacelnik, 2010*; *Pompilio et al., 2006*; *Solvi et al., 2022*; *Matsumoto and Mizunami, 2004*; *Ferrucci et al., 2019*).

On the other side, by contrasting binary to trinary options decision problems, we were able to provide robust evidence against the idea that value context-dependence follows the functional form of divisive normalization in the reinforcement learning scenario (*Louie and Glimcher, 2012*; *Louie et al., 2015*; *Louie, 2022*; *Glimcher, 2022*; *Daviet and Webb, 2023*). Our demonstration relied on the straightforward and well-accepted idea that virtually any instantiation of the divisive normalization model would predict a strong (we specify 'strong' because a minimal non-null choice set size effect is also predicted by simply assuming choices deriving from a softmax decision rule) choice (description-based task) or outcome (reinforcement-based task) set size effect: all things being equal, the subjective value of an option or an outcome in a trinary decision context should be lower compared to that of a similar value belonging to a binary context. We find no evidence for such an effect. In fact, if anything, the subjective values of options belonging to trinary decision contexts were numerically higher compared to that of the binary decision contexts. Beyond the quantitative analysis of behavioral performance, model fitting and simulations analysis also revealed that the divisive normalization model dramatically failed to correctly reproduce the behavioral pattern (*Palminteri et al., 2017*). Crucially, this was also true for fully parameterized versions of the divisive normalization model (*Webb et al., 2021*; *Figure 3—figure supplement 1*).

The behavioral results were rather consistent with an alternative rule, range normalization, according to which subjective value signals are normalized as a function of the maximum and minimum values (regardless of the number of options). This normalization rule is reminiscent of the range principle proposed by Parducci to explain perceptual (and later also affective) judgments (*Parducci, 1995*). In contrast to divisive normalization, range normalization predicts that contextually high- and low-value

options are assigned the same subjective value regardless of their absolute value and the number of options in a given choice set. This behavior hallmark of divisive normalization was falsified in all experiments. Of note, although the quality of the fit of the range normalization model was significantly better compared to that of the divisive normalization model, in terms of both predictive accuracy (out-of-sample likelihood) and behavioral signatures, it manifestly failed to properly capture the subjective values attributed to the mid-value options in trinary choice contexts. This pattern was not affected by introducing forced-choice trials to focus the participants' attention on mid-option outcomes (for at least five trials) (*Chambon et al., 2020*). To further improve the range normalization model fit, we endowed it with a nonlinear weighting of normalized subjective values. Nonlinear weighting parameters, although they do not provide mechanistic accounts, are often introduced in models of decision-making to satisfactorily account for utility and probability distortions (*Hertwig and Erev, 2009*; *Wakker, 2010*). Introducing this weighting parameter improved the model fit qualitatively and quantitatively in a substantial manner. Importantly, introducing the same parameter into the divisive model did not improve its performance significantly (*Figure 3—figure supplement 1*). The weighted range normalization model improved the fit assuming that the mid- and low-value options are subjectively perceived as much closer than they are in reality. We believe that this may derive from attentional mechanisms that bias evidence accumulation as a function of outcomes and option expected values (*Spektor et al., 2019*; *Krajbich et al., 2010*; *Zilker and Pachur, 2022*). To further probe this hypothesis, we designed and ran an additional experiment (Experiment 3) where we manipulate the possibility of choosing the high-value option in trinary learning contexts. This manipulation successfully managed to 'correct' the subjective valuation of mid-value options (while leaving unaffected the valuation of the other options). The behavior in this experiment was successfully captured by further tweaking the weighted range normalization model by assuming that different weighting parameters apply to the chosen and unchosen outcomes. By finding concave and convex weighting functions for the chosen and unchosen outcomes, respectively, the model managed to explain why forcing the participant to choose the mid-value option increases its subjective valuation. We believe that these results further reinforce the hypothesis that outcome valuation interacts with attentional deployment. In fact, it is reasonable to assume that the obtained outcome is attended more than the forgone ones (after all it is the more behaviorally relevant outcome) and that increased attention devoted to the obtained outcomes "boosts" its value (*Krajbich et al., 2010*). This effect can also be conceptually linked to a form of choice-confirmation bias, where the mid-value outcome is perceived as better than it actually is (*Chambon et al., 2020*).

In addition to assessing subjective values using choices as standardly done in behavioral economics and nonhuman animal-based neuroscience, we also assessed subjective option values using explicit ratings (*Garcia et al., 2021*). Despite the fact that a wealth of evidence in decision-making research suggests that subjective values are highly dependent on whether they are inferred from choices or ratings (also referred to as the *revealed* versus *stated* preferences dichotomy; *Lichtenstein and Slovic, 2006*), post-learning explicit ratings delivered results remarkably consistent with choice-revealed preferences. Indeed, transfer phase choices could almost perfectly be reproduced from explicit ratings, which were, in turn, more consistent with range, rather than divisive normalization process. In addition to provide a welcome test of robustness of our results, the similarity between choice-based and rating-based subjective values also demonstrates the context-dependent valuation span across procedural as well as declarative representational systems (*Gershman and Daw, 2017*; *Biderman and Shohamy, 2021*).

Beyond the specific question of its functional form, one could ask why option values would be learned and represented in a relative or normalized manner? In other terms, what is the functional reason for context-dependent representations? One partial answer to this question can be tracked to studies showing that context-dependent preferences are ecologically rational (in other words, they convey a statistical or strategical advantage over unbiased representations; *McNamara et al., 2012*). In a similar vein, it could be proposed that unbiased value representations are computationally more costly, making relative or context-dependent encoding an efficient solution. Consistent with this idea, a recent study indicates that human participants are capable of adaptively modulating their value representations from relative to absolute as they learn that the latter scheme is more advantageous (*Juechems et al., 2022*). Another study confirmed that value representations are somehow flexible by showing that shifting the attentional focus from subject emotions to perceive outcome shifts the

balance from relative to unbiased value encoding (*Hayes and Wedell, 2022*). However, neither of these studies manage to report situations in which the representational code was fully context-independent.

Taking into account that relative value encoding has been shown in a plethora of species and situations (*Yamada et al., 2018*; *Conen and Padoa-Schioppa, 2019*; *Pompilio and Kacelnik, 2010*; *Pompilio et al., 2006*; *Solvi et al., 2022*; *Matsumoto and Mizunami, 2004*; *Ferrucci et al., 2019*), it seems also reasonable to suppose that these findings stem from some deep preserved principles of how (value-based) information is encoded and represented in the brain. Accordingly, normalization naturally emerges as a solution to maximize the gain function between underlying stimuli (whose range may vary greatly) and neural response (*Barlow, 1961*). Crucially, and consistently with our results, while there is ample evidence that the divisive normalization rule provides a good account of information encoding in the perceptual system (*Carandini and Heeger, 2011*), several primate neurophysiological studies indicate the value-related signals in dopaminergic neurons (*Tobler et al., 2005*) and the orbitofrontal cortex (*Padoa-Schioppa, 2009*; *Kobayashi et al., 2010*): hubs of the brain valuation system (*Bartra et al., 2013*; *Pessiglione and Delgado, 2015*). Similar findings have been replicated in human fMRI (*Burke et al., 2016*; *Cox and Kable, 2014*; *Pischedda et al., 2020*). On the other side, neural evidence of divisive normalization in value-based decision-making is scant in the brain valuation system (but see *Yamada et al., 2018*), although it remains possible that activity in the perceptual and attentional systems (such as the parietal cortex) displays signs of divisive normalization (*Louie et al., 2011*).

Multiple elements of our results concordantly show that divisive normalization does not provide a good account of subjective value representation in human reinforcement learning. More precisely, we were concerned about whether at the outcome stages the subjective values of rewards were normalized according to a divisive (or range) normalization rule (*Juechems et al., 2022*; *Louie, 2022*). It is nonetheless still possible that this rule provides a good description of human behavior in other value-based decision-making domains. In fact, most of the previous studies claiming evidence for divisive normalization used other tasks involving items whose values are described (such as snacks or lotteries food) and have not to be extracted from experience (*Louie et al., 2013*; *Webb et al., 2021*; *Bucher and Brandenburger, 2022*; *Robinson and Tymula, 2019*). In addition, our study only addressed value normalization of the outcome magnitude space and did not address whether the same rule would apply to other outcome attributes, such as probability (*Daviet and Webb, 2023*).

However, it is worth noting that evidence concerning previous reports of divisive normalization in humans has been recently challenged and alternative accounts, such as range normalization, have not been systematically tested in these datasets (*Gluth et al., 2020*; *Webb et al., 2020*, although see *Webb et al., 2021* for an exception, where nonetheless the range and the divisive models were not given equal chances due to different parametrization). It is worth mentioning that the range normalization principle has been recently successfully adapted to account for decision-making under risk (*Kontek and Lewandowski, 2018*). On the other side, another recent study compared range and divisive normalization in multi-attribute, description-based decision problems and found evidence for divisive normalization (*Daviet and Webb, 2023*). Taken together, we believe that our and other recent findings call for a critical appraisal of normalization in value-based decision-making comparing with alternative models and using highly diagnostic experimental designs, as the one used here.

On the other side, by using a reinforcement learning framework, our design has the advantage that it can be readily translated into animal research to further extend and characterize the neural mechanisms underlying these findings. Further research will also determine whether the range normalization rule also applies to primary (positive and negative) rewards even if previous evidence (in humans and animals) suggests that context-dependent principles apply to food and electric shocks (*Yamada et al., 2018*; *Conen and Padoa-Schioppa, 2019*; *Pompilio and Kacelnik, 2010*; *Pompilio et al., 2006*; *Solvi et al., 2022*; *Matsumoto and Mizunami, 2004*; *Ferrucci et al., 2019*; *Vlaev et al., 2009*).

Despite the fact that an even a 'naïve' range normalization presents several computational advantages compared to divisive normalization (such as the fact of being easily translatable to partial feedback and punishment avoidance tasks; *Bavard et al., 2018*; *Bavard et al., 2021*), we also showed that its fit was far from perfect and there was still room for improvement. For instance, the weighted range normalization rule, although descriptively accurate, is silent concerning its cognitive origin mechanism. Future research, for example, featuring eye-tracking, will be necessary to shed light on these aspects. Future research will also be needed to assess the extent to which the same rule applies to

vast decision spaces involving more than three options. Finally, further experiments will be needed to generalize the current models to partial feedback situations where the contextual variables have to be inferred and stored in memory.

Despite the fact that including the (attentional) weighting parameter improved the quality of fit (both in terms of out-of-sample log-likelihood and model simulations) of range normalization process, we acknowledge that some features of the data were still not perfectly accounted. For instance, even if the effect was small in size, from averaging across several experiments it appeared that the choice rate for the high-value options in the trinary contexts was higher compared to those in the binary ones. Although this feature provides strong evidence against the divisive normalization framework (which predicts the opposite effect), it is also not coherent with the range normalization process. It could be hypothesized that other cognitive and valuation mechanisms concur to generate this effect, such as instance-based or comparison-based decision valuation processes where the options in the trinary contexts would benefit from an additional (positive) comparison (*Gonzalez and Lebiere, 2005*; *Vlaev et al., 2011*).

To conclude, while our results cast serious doubt about the relevance of the divisive normalization principle in value-based decision-making (*Glimcher, 2022*), they also establish once again that context-dependence represents one of the most pervasive features of human cognition and provides further insights into its algorithmic instantiation.

## Materials and methods
### Participants
Across three experiments, we recruited 500 participants (227 females, 243 males, 30 N/A, aged 26.44 ± 8.31 years old) via the Prolific platform (https://www.prolific.co). The research was carried out following the principles and guidelines for experiments including human participants provided in the Declaration of Helsinki (1964, revised in 2013). The INSERM Ethical Review Committee/IRB00003888 approved the study, and participants were provided written informed consent prior to their inclusion. The results presented in the main text are those of Experiment 1 (N = 150) and Experiment 3 (N = 200). The results of an alternative design (Experiment 2) are presented in *Figure 2—figure supplement 2* and *Figure 6—figure supplement 1*. To sustain motivation throughout the experiment, participants were given a bonus depending on the number of points won in the experiment (average money won in pounds: 5.05 ± 0.50, average performance against chance during the learning phase and transfer phase: M = 0.74 ± 0.087, $t(149) = 34.04$, p<0.0001, $d = 2.78$). The data of one participant for the explicit phase was not included due to technical issues. A pilot online-based experiment was originally performed (N = 40, the results are also presented in *Figure 2—figure supplement 1*).

### Behavioral tasks
Participants performed an online version of a probabilistic instrumental learning task, instantiated as a multiarmed bandit task. After checking the consent form, participants received written instructions explaining how the task worked and that their final payoff would be affected by their choices in the task. During the instructions, the possible outcomes in points (from 0 to 100 points) were explicitly showed as well as their conversion rate (1 point = 0.02 pence). The instructions were concluded with a short three-item quiz to make sure participants' understanding of the task was sufficient. The instructions were then followed by a short training session of 12 trials aiming at familiarizing the participants with the response modalities. If participants' performance during the training session did not reach 60% of correct answers (i.e., choices toward the option with the highest expected value), they had to repeat the training session. Participants could also voluntarily repeat the training session up to two times and then started the actual experiment.

In our main task, options were materialized by abstract stimuli (cues) taken from randomly generated identicons, colored such that the subjective hue and saturation were very similar according to the HSLUV color scheme (https://www.hsluv.org). On each trial, two or three cues were presented on different positions (left/middle/right) on the screen. The position of a given cue was randomized, such that a given cue was presented an equal number of times on the left, the middle, and the right. Participants were required to select between the cues by clicking on one cue. The choice window was

self-paced. A brief delay after the choice was recorded (500 ms); the outcome was displayed for 1000 ms. There was no fixation screen between trials.

### Experimental design (version a)

The full task consisted of three phases: one learning phase and two elicitation phases. During the learning phase, cues appeared in fixed pairs/triplets. Each pair/triplet was presented 45 times, leading to a total of 180 trials. Within each pair/triplet, the cues were associated to a deterministic outcome drawn from a normal distribution with variable means $\mu \epsilon \left[0, 100\right]$ and fixed variance $\sigma = 4$ (*Table 1*). At the end of the trial, the cues disappeared and were replaced by the outcome. Once they had completed the learning phase, participants were displayed with the total points earned and their monetary equivalent.

After the learning phase, participants performed two elicitation phases: a transfer phase and an explicit rating phase. The order of the elicitation phases was counterbalanced across participants. In the transfer phase, the 10 cues from the learning phase were presented in all possible binary combinations (45, not including pairs formed by the same cue). Each pair of cues was presented four times, leading to a total of 180 trials. Participants were explained that they would be presented with pairs of cues taken from the learning phase, and that all pairs would not have been necessarily displayed together before. On each trial, they had to indicate which of the cues was the one with the highest value. In the explicit rating phase, each cue from the learning phase was presented alone. Participants were asked what was the average value of the cue and had to move a cursor ranging from 0 to 100. Each cue was presented four times, leading to a total of 40 trials. In both elicitation phases, the outcome was not provided in order not to modify the subjective option values learned during the learning phase, but participants were informed that their choices would count for the final payoff.

### Experimental design (versions b and c)

In the learning phase, we added forced-choice trials to the 180 free-choice trials (*Chambon et al., 2020*). In these forced trials, only one option was selectable and the other cue(s) were shaded. We added five forced-choice trials per option, leading to a total of 230 trials in the learning phase. In version b, even in the forced-choice trial, the participants could only see the outcomes of all options. In version c, participants could only see the outcome of the chosen option. The elicitation phases (transfer and explicit rating) remained unchanged.

### Experiment 3

In the learning phase, cues appeared in fixed triplets only. Each triplet was presented 45 times, leading to a total of 180 trials. We used a 2 × 2 design manipulating the range spread (as in Experiment 1a) and the option availability: in half of the contexts, for some proportion of trials (Experiment 3a: 50%; Experiment 3b: 75%), the most favorable option was unavailable (*Figure 5A*). It was displayed on the screen with a shaded mask and was not clickable. At the end of each trial, all cues disappeared and were replaced by the outcome (shaded outcome for the nonclickable option). In the transfer phase, the 12 cues from the learning phase were presented in all possible binary combinations (66, not including pairs formed by the same cue). Each pair of cues was presented two times, leading to a total of 132 trials. In the explicit phase, each cue was presented two times, leading to a total of 24 trials.

## Behavioral analyses

For all experiments, we were interested in three different variables reflecting participants' learning: (1) correct choice rate in the learning phase, that is, choices directed toward the option with the highest objective value; (2) choice rate in the transfer phase, that is, the number of times an option is chosen, divided by the number of times the option is presented; and (3) subjective valuation in the explicit phase, that is, average reported value per option. Statistical effects were assessed using multiple-way repeated-measures ANOVAs with range amplitude (narrow or wide) and number of presented options (binary or trinary decision problem, *Figure 1*) as within-participant factor and experiment version (a,b,c) as between-participant factors. Post hoc tests were performed using one-sample and two-sample *t*-tests for respectively within- and between-experiment comparisons. To assess overall performance, additional one-sample *t*-tests were performed against chance level (0.5 – two-option contexts – and 0.33 – three-option contexts). We report the *t*-statistic, p-value, and Cohen's *d* to estimate

effect size (two-sample $t$-test only). Given the large sample size (n = 500), central limit theorem allows us to assume normal distribution of our overall performance data and apply properties of normal distribution in our statistical analyses, as well as sphericity hypotheses. Concerning ANOVA, we report Levene's test for homogeneity of variance, the uncorrected statistical, as well as Huynh–Feldt correction for repeated-measures ANOVA when applicable (**Girden, 1992**), $F$-statistic, p-value, partial eta-squared ($\eta_p^2$), and generalized eta-squared ($\eta^2$) (when Huynh–Feldt correction is applied) to estimate effect size. All statistical analyses were performed using MATLAB (https://www.mathworks.com) and R (https://www.r-project.org).

## Computational models

The goal of our models is to estimate the subjective value of each option and choose the option that maximizes the expected reward (in our case, with the highest expected value). At each trial $t$, in each context $s$, the expected value $Q$ of each option $i$ is updated with a delta rule:

$$Q_t(s, i) = Q_{t-1}(s, i) + \alpha_X * \delta_t \tag{4}$$

where $\alpha_X$ is the learning rate and $\delta_t$ is a prediction error term. For all our models, at each trial, the chosen and unchosen options are updated with two distinct learning rates for chosen ($\alpha_C$) and unchosen ($\alpha_U$) options, and separate, outcome-specific, prediction error terms $\delta_t$, calculated as the difference between the subjective outcome $u(R_i)$ and the expected one:

$$\delta_t = u(R_i) - Q_{t-1}(s, i) \tag{5}$$

We modeled participants' choice behavior using a softmax decision rule representing the probability for a participant to choose one option $a$ over the other options – one alternative in binary contexts ($n = 2$), two in trinary contexts ($n = 3$) in each context $s$:

$$P_t(s, a) = \frac{e^{Q_t(s,a) * \beta}}{\sum_{k=1}^{n} e^{Q_t(s,k) * \beta}} \tag{6}$$

where $n$ is the number of outcomes presented in a given trial ($n = 2; n = 3$) and $\beta > 0$ is the inverse temperature parameter. High temperatures ($\beta \to 0$) cause the action to be all (nearly) equiprobable. Low temperatures ($\beta \to +\infty$) cause a greater difference in selection probability for actions that differ in their value estimates (**Sutton and Barto, 1998**).

We compared four alternative computational models: the unbiased (UNBIASED) model, which encodes outcomes on an absolute scale independently of the choice context in which they are presented; the range normalization (RANGE) model, where the reward is normalized as a function of the range of the outcomes, the divisive normalization (DIVISIVE) model, where the reward is normalized as a function of the sum of all the outcomes; and the nonlinear range normalization (RANGE$^\omega$) model, where the normalized outcome is power-transformed with an additional free parameter.

### Unbiased model

At trial $t = 0$, for all contexts $Q_{t=0} = 50$. For each option $i$, the subjective values $u(R_i)$ are encoded as the participants see the outcomes, that is, their objective value in points.

$$u(R_i) = R_i, R_i \in [0, 100] \tag{7}$$

### Range normalization model

At trial $t = 0$, for all contexts $Q_{t=0} = 0.5$. The subjective values $u(R_i)$ are encoded depending on the value of the other options, specifically the maximum and the minimum available rewards.

$$u(R_i) = \frac{R_i - min(R_:)}{max(R_:) - min(R_:)} \tag{8}$$

where $max(R_:)$ and $min(R_:)$ are, respectively, the maximum and minimum outcomes presented in a given trial. In version c, where only the reward of the chosen option is displayed, the outcomes of unchosen options are replaced with the last seen outcomes for these options (**Spektor et al., 2019**).

## Divisive normalization model

At trial $t = 0$, for all options $Q_{t=0} = 0.5$. The outcomes are encoded depending on the value of all the other options, specifically the sum of all available rewards.

$$u\left(R_i\right) = \frac{R_i}{\sum_{k=1}^{n} R_k} \qquad (9)$$

where $n$ is the number of outcomes presented in a given trial ($n = 2; n = 3$). In version c, where only the reward of the chosen option is displayed, the outcomes of unchosen options are replaced with the last seen outcomes for these options (*Spektor et al., 2019*).

## Nonlinear range normalization model

At trial $t = 0$, for all contexts $Q_{t=0} = 0.5$. The subjective values $u\left(R\right)$ are encoded depending on the value of the other options, specifically the maximum and the minimum available rewards. This normalized outcome is then set to the power of $\omega$, with $0 < \omega < +\infty$:

$$u\left(R_i\right) = \left(\frac{R_i - min(R_:)}{max(R_:) - min(R_:)}\right)^{\omega} \qquad (10)$$

where $max\left(R_:\right)$ and $min\left(R_:\right)$ are, respectively, the maximum and minimum outcomes presented in a given trial. In version c, where only the reward of the chosen option is displayed, the outcomes of unchosen options are replaced with the last seen outcomes for these options (*Spektor et al., 2019*).

## Conditional, nonlinear range normalization model

Finally, in Experiments 3a and 3b only, we tested a more complex version of the model, which allowed for different weighting parameters for obtained ($\omega_c$) and forgone ($\omega_u$) outcomes. The weighting parameters were allowed same range as before.

## Ex ante simulations

The model predictions displayed in *Figure 1C* were obtained by simulating choices of artificial agents. The simulated choices were equivalent to those later performed by the participants, that is, 180 trials (45 per learning contexts) in the learning phase (where the deterministic outcomes were drawn from a normal distribution with variable means $\mu \epsilon \left[0, 100\right]$ and fixed variance $\sigma = 4$) and 180 trials (4 per comparison) in the transfer phase. The update rule for the option values is described in *Equations 1 and 2*. Predictions for each experiment were simulated for a set of 50 agents to match our number of participants per version. Each agent was associated with a set of parameters $\left[\beta, \alpha|c, \alpha_u\right]$ for the inverse temperature, the learning rate of the chosen option, and the learning rate for unchosen options, respectively. The parameters were independently drawn from prior distributions, which we took to be Beta(1.1,1.1) for the learning rates and Gamma(1.2,5) for the inverse temperature (*Daw et al., 2011*). The value of the inverse temperature is irrelevant in the learning phase because the feedback is always complete, which means that the options should converge, in average, to their subjective average value independently of the choice, provided that the learning rates are different from 0. Moreover, in the transfer phase, to obtain the agents' preferences based on the learned option values, we chose to use an argmax decision rule instead of a softmax decision rule. At each trial $t$ in the transfer phase, comparing option $a$ and option $b$, the probability of choosing option $a$ is calculated as follows:

$$\mathrm{P}_t\left(a\right) = \begin{cases} 1 & if\ Q_f\left(a\right) > Q_f\left(b\right) \\ 0.5 & if\ Q_f\left(a\right) = Q_f\left(b\right) \\ 0 & if\ Q_f\left(a\right) < Q_f\left(b\right) \end{cases} \qquad (11)$$

where $Q_f$ is a vector of the final $Q$-values at the end of the learning phase.

## Acknowledgements

SP is supported by the European Research Council under the European Union's Horizon 2020 research and innovation program (ERC) (RaReMem: 101043804), and the Agence Nationale de la Recherche

(CogFinAgent: ANR-21-CE23-0002-02; RELATIVE: ANR-21-CE37-0008-01; RANGE: ANR-21-CE28-0024-01). The Departement d'études cognitives is supported by the Agence National de la Recherche (ANR; FrontCog ANR-17-EURE-0017). SB acknowledges support from the European Research Council (ERC) under the European Union's Horizon 2020 research and innovation program (grant agreement no. 948545).

## Additional information

### Funding

| Funder | Grant reference number | Author |
|---|---|---|
| European Research Council | 101043804 | Stefano Palminteri |
| Agence Nationale de la Recherche | ANR-21-CE23-0002-02 | Stefano Palminteri |
| Agence Nationale de la Recherche | ANR-21-CE37-0008-01 | Stefano Palminteri |
| Agence Nationale de la Recherche | ANR-21-CE28-0024-01 | Stefano Palminteri |

The funders had no role in study design, data collection and interpretation, or the decision to submit the work for publication.

### Author contributions

Sophie Bavard, Conceptualization, Data curation, Formal analysis, Investigation, Visualization, Methodology, Writing – original draft, Writing – review and editing; Stefano Palminteri, Conceptualization, Supervision, Funding acquisition, Validation, Investigation, Visualization, Methodology, Writing – original draft, Project administration, Writing – review and editing

### Author ORCIDs

Sophie Bavard (iD) http://orcid.org/0000-0002-9283-2976
Stefano Palminteri (iD) http://orcid.org/0000-0001-5768-6646

### Ethics

The research was carried out following the principles and guidelines for experiments including human participants provided in the declaration of Helsinki (1964, revised in 2013). The INSERM Ethical Review Committee / IRB00003888 approved and participants were provided written informed consent prior to their inclusion.

### Decision letter and Author response

Decision letter https://doi.org/10.7554/eLife.83891.sa1
Author response https://doi.org/10.7554/eLife.83891.sa2

## Additional files

### Supplementary files

• MDAR checklist

### Data availability

Data and codes are available here https://github.com/hrl-team/3options (copy archived at *Bavard and Palminteri, 2023*).

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
