## [Editor Report]

This important study presents a series of behavioral experiments that test whether value normalization during reinforcement learning follows divisive or range normalization. Behavioral data from probe tests with two alternatives demonstrate convincingly that range normalization provides a better account for choice behavior and value ratings in this setting. These findings will be of interest for readers interested in neuroeconomics and cognitive neuroscience.

---

## [Decision Letter]

**Decision letter after peer review:**

Thank you for submitting your article "The functional form of value normalization in human reinforcement learning" for consideration by *eLife*. Your article has been reviewed by 3 peer reviewers, and the evaluation has been overseen by a Reviewing Editor and Michael Frank as the Senior Editor. The following individual involved in the review of your submission has agreed to reveal their identity: Neil Garrett (Reviewer #3).

Essential revisions:

As you can see in the individual comments there was some confusion about key aspects of your experimental approach and measures. This confusion was resolved during a discussion among reviewers. It would be important that your revised manuscript is more clear about these key aspects. In addition, it would be important to clearly discuss the differences between the predictions made by the different models in choice and learning contexts.

*Reviewer #1 (Recommendations for the authors):*

1. It would be important to more clearly (and early on) state how 'choice rate' was computed for the transfer phase. This may require describing the transfer phase in more detail.

2. In lines 272 and 299, please don't use 'perfectly' to qualify the match between model predictions and behavior. I agree that the match is pretty close, but perfectly implies an errorless match, which is not the case here. The authors could consider using 'closely captures' and 'closely match' instead.

3. There are several typographical errors throughout the manuscript. I am listing a few below but I'd encourage the authors to proofread the manuscript.

– Line 190: "WB_86 and WT_86", not "NB_86 and NT_86".

– Line 367-368: parenthesis is not closed.

– Line 377-378: parenthesis is not closed.

– Line 382: "that value representations are" not "that values representations are".

*Reviewer #2 (Recommendations for the authors):*

The main issues with the paper are expositional. Both a clarification of the simulations used to generate the predictions (Figure 1C) and a clarification of the predictions and appropriate setting of each model in previous literature is needed. In particular, DN makes no relevant predictions in a binary choice setting, so in my initial ignorance, I simply assumed the transfer phase was a trinary choice (because otherwise, DM has no bite). I was so confused by these issues that I needed a discussion with the editor after I submitted my initial report. I have left these comments in my report below (points 1-3 below) so the authors are aware of this miscommunication and that it might help them improve Figure 1 and lines 110 – 130. Finally, the statement of the results (and discussion) is rather opinionated. The writing could do with more nuance regarding the predictions of each model, the appropriate setting for those predictions, and the patterns in the data. In particular, there appear to be patterns that also reject a range normalization account, suggesting that we still have much to learn on this topic.

Before discussion with the editor:

1) The choice probabilities do not sum to 1 within condition (for example, in treatment NB, the "50" option seems to be chosen with probability.66 and the "14" with probability <.2. Similar issues arise for DIVISIVE and RANGE. Either I deeply misunderstand the experimental setting, or there is something wrong with the simulation underlying the figure (the simulation uses an argmax, so I do not see how the probabilities can not sum to 1).

2) Why are the unbiased choice probabilities for "50" equal across NB and NT in UNBIASED? Loosely speaking, the Q-learning rule used by the authors will generate a Q_45 that is normally distributed around means [14,50,32,86] (see SuttonandBarto Equation 2.6 and impose a CentralLimitTheorem). Taking the arg-max of these normal random variables will require that P(50) (weakly) decreases as the set size increases. The authors seem to be claiming that it is constant.

3) In an effort to find the causes of these differences, I attempted to simulate the model. Apart from a number of issues with the description of the computational model (see below), the reported simulation size leads to substantial variation between simulation runs (with different RNG seeds).

– Note that I did not include heterogeneity in the learning rate in these simulations, since the theoretical setting does not require it (at least the authors don't claim that it does). So α is set to 0.5, which is the mean of the Β(1.1,1.1) distribution the authors use.

After discussion with the editor:

5) Line 123: "It is calculated across all the comparisons involving a given option." This seemingly innocuous sentence needs much more explanation. Does the transfer phase include ALL binary choices between ALL stimuli, including those across conditions? So is the "86" from the WB paired with the "86" from "WT"? If so, presumably this is what is generating the identical choice rates in UNBIASED since they are essentially the same Q_45(86). It is still deeply unsettling that nothing adds up to 1 in Figure 1C, but at least now I understand the source of the differing predictions. However, there still are some issues:

– Why is there a slight increase in the choice proportion for 86 between WT and WB? Shouldn't Q_45 be the same for them because the range doesn't change?

– The number of simulations should be increased (i.e. is it just "10 choose 2" x 50 choices). Given comment 3 above, are we sure there is no noise in the predictions (see points above)? Also, the issue of introducing α heterogeneity into the simulation needs to be addressed, as it seems pointless.

6) The framing in the introduction of the paper needs to be clear about the use of a binary choice transfer phase. DN, as conceptualized by previous work the authors cite, is a model for how valuations are normalized in the choice process (e.g. Webb et al., Mng Sci, 2020, Figure A.4). As such, it is acknowledged that it has limited prediction in binary choice (see Webb et al., Mng Sci, 2020 Proposition 1). Nearly all of its testable implications arise in trinary (or larger) choice sets. For example, the quote in Footnote 1 is the first sentence of the first paragraph of a section titled "Choice Set Size Effects" which describes studies in which there are more than two alternatives in the choice set. There is no choice set size effects in this experiment because it only examines binary choice in the transfer phase, therefore it excludes the types of choice sets in which DN has been previously observed.

As such, the authors write that they opt for a reinforcement learning paradigm "because it has greater potential for translational and cross-species research". It may indeed have translational benefits for cross-species research, and learning settings are important in their own right, but the use of an RL paradigm is not innocuous. There may be critical differences in the form of value coding computations between conditioned learning settings and "descriptive" settings in which valuations are already known (as acknowledged by the authors in an earlier ScienceAdvances paper), not the least of which might be due to the differences between cortical and sub-cortical neural computations (the neural evidence for DN is primarily in cortical regions). There is only one paper I am aware of that applies DN in an RL setting (Louie, PLOSCompBio2022) to account for gain/loss asymmetry in behaviour without the need for different learning rates (though not cited by the authors). This paper should make clear that the application of DN to a conditioned learning phase is a hypothesis raised in this paper, and clarify the distinction between settings in which DN has been previously proposed and examined in other work.

7) Previous studies add alternatives to a choice set and study value normalization. They are important distinctions from this work because they test for choice set size effects explicitly. Exp 2 in Webb et al. 2020 does such an experiment (albeit not in a learning setting) and finds support for DN rather than RN. The key difference is that the additional alternatives are added to the choice set, rather than only in a learning phase. Daviet and Webb also report a double decoy experiment of description-based choice, which follows a similar intuition to this experiment by placing an alternative "inside" two existing alternatives (though again, directly in the choice set rather than a learning phase). The conclusions from that paper are also in favour of DN rather than RN since this interior alternative appears to alter choice behaviour. This latter paper is particularly interesting because it relies on the same intuition for experimental design as in the experiment reported here. The difference is just the learning vs. choice setting, suggesting this distinction is critical for observing predictions consistent with RN vs DN.

8) The experiment tests RN vs DN by expanding the range upward with the "86" option. However, RN would make an equivalent set of predictions if the range were equivalently adjusted downward instead (for example by adding a "68" option to "50" and "86", and then comparing to WB and WT, because it effectively sets the value to "0" or "1" depending only on the range). The predictions of DN would differ however because adding a low-value alternative to the normalization would not change it much. Given the design of the experiment, this would seem like a fairly straightforward prediction to test for RN. Would the behaviour of subjects be symmetric for equivalent ranges, as RN predicts? If so this would be a compelling result, because symmetry is a strong theoretical assumption.

9) There were issues with the reporting of the results that need to be addressed:

line 183, typo NB_50?

lines 182-188 need copy-editing.

line 186; "numerically lower" has no meaning in a statistical comparison. The standard error (t-test) is telling us that there is too much uncertainty to determine if there is a difference between treatments. No more, no less. Arguing for "numerical" significance is equivalent to arguing that there is signal in noise. Statements like this should be removed from the paper.

line 188 – "superficially"? a test either rejects a prediction or doesn't. More specifically, this observation also rejects a prediction of RN and should be stated as such for clarity.

line 183-184 – given that experiments a, b, and c, seem similar and are pooled in later analyses, it would be useful to also pool them to test whether there is an increase in the NT_86 vs NB_86 and WT_86 vs WB_86 choice rates (perhaps using a pooled paired t-test). A tripling of the number of observations might be helpful here for testing whether NB_50<nt_50<wb_86

line 272 – "perfectly" is an overstatement, given the log-likelihood is not zero. Also see the NB_50<nt_50<wb_86

line 299 – perhaps also not "perfect" What is the t-test reported here? And how to interpret this p-value of.13? Is this an argument for accepting the null hypothesis?

*Reviewer #3 (Recommendations for the authors):*

I have a few comments for the authors to consider:

1) Training performance (Figure 2) seems very close to the ceiling (i.e. accuracy of 100%) for many of the participants. I would be slightly worried that there isn't enough stochasticity for some participants to derive meaningful parameters from the model – did the authors worry about this or run any checks to guard against this possibility at all? Perhaps more interestingly, this pattern (which suggests participants quickly identify the best option and just keep selecting this) also made me wonder whether the unbiased model with a persistence parameter (i.e. which bonuses the most recent choice) would fare as well in terms of log-likelihood scores (Table 2) and/or capture the same pattern of probe (transfer phase) choices as either the divisive or range models. Could the authors refute this possibility?

2) Previous models from this group (e.g., Palminteri 2015 Nature Comms) have conceptualised "contextual value" as an iteratively learned quantity (like the Q values here). But in the normalisation models presented, this aspect is absent. For example, min(R) and max(R) in equation 8 used for the range normalisation model are simply the min/max rewards on that trial (or on the previous trial when full feedback is absent). Similarly, the denominator in equation 9 for the divisive normalisation model. In practice, these could each be updated. For example, instead of min(Ri) in eq. 9, they could use something along the lines of Vmin(i) which updates as:

Vmin(i)t+1 = Vmin(i)t + α*δ t

δt = min(Ri) – Vmin(i)t

Did the authors consider this? Or, another way of asking this, is to ask why the authors assume these key components of the model are not learned quantities that get updated over trial-by-trial experience.

3) Transfer phase – the transfer phase only contains binary combinations. However, during training participants were presented with two binary combination choices and two trinary combination choices (i.e. choices were 50% binary choices and 50% trinary). Could exclusively using binary combinations in the probe phase bias the results at all towards finding stronger evidence for the range model? I'm thinking (and I could be wrong!) that the divisive approach might be better suited to a world where choices are over >2 options. So perhaps participants make choices more consistent with the range model when the probe phase consists of binary choices but might switch to making choices more consistent with the divisive model if this consisted of trinary choices instead. The explicit rating (Figure 5) results do go some way to showing that the results are not just the result of the binary nature of the probe phase (as here ratings are provided one by one I believe). Nonetheless, I'd be interested if the authors had considered this at all or considered this a limitation of the design used. </nt_50<wb_86</nt_50<wb_86

[Editors' note: further revisions were suggested prior to acceptance, as described below.]

Thank you for resubmitting your work entitled "The functional form of value normalization in human reinforcement learning" for further consideration by *eLife*. Your revised article has been evaluated by Michael Frank (Senior Editor) and a Reviewing Editor.

The manuscript has been improved but there are some remaining issues that need to be addressed, as outlined below:

As you can see, two of the original reviewers were satisfied with your revisions. However, Reviewer #2 does not feel some of their points were fully addressed. Most importantly, there are open issues regarding the contextualization of the current study as well as the important fact that DN was designed to account for choice set effects, rather than normalization during learning, as tested here. Specifically, the binary choice test is only sensitive to value normalization during learning but not whether the form or normalization of these learned values during choice. In other words, while the current experiments test the interesting and worthwhile question of how values are normalized during learning, they do not test whether DN is still applied to RN-learned values at the time of choice. In summary, there is an important distinction between the design of the current study and what has been done to study DN in the past as well as how DN was originally conceptualized. We agree with Reviewer #2 and feel that this distinction should be acknowledged and clarified throughout the paper, most importantly early on in the introduction. Also, the discussion would benefit from revisions that restrict the conclusions of the findings to the appropriate (i.e., learning) context. We are looking forward to receiving a revised version of your interesting and important manuscript.

*Reviewer #2 (Recommendations for the authors):*

The authors have addressed some of my comments from the previous round, though a few remain. In particular, there still seems to be some disagreement over how to describe the study and place it in the context of previous literature. I realize that the authors are addressing (what they interpret) to be overly-strong general statements about the form of normalization in decision-making. This is a useful undertaking. But repeating the same mistake of over-generalizing and not placing the results in context will only serve to continue the confusion.

1) In their reply, the authors write:

"The outcome presentation stage is precisely where (range or divisive) normalization is supposed to happen and affect the values of the options (see also Louie et al., 2022). Once values have been acquired during the learning phase (and affected by a normalization process that may or may not depend on the number of outcomes), it does not matter if they are tested in a binary way."

and:

"First of all, we note that nowhere in the papers proposing and/or defending the DN hypothesis of value-based choices, we could find an explicit disclaimer and/or limitation indicating the authors believe that it should only apply to prospective valuation of described prospects and not to retrospective valuation of obtained outcomes."

I fundamentally disagree on this point. There are various opinions on where normalization is "supposed to happen." Multiple papers have conceptualized DN as a process that occurs during decision and not learning (the citations were provided in the previous round, including some that I feel qualified to interpret). Louie et al., 2022 is the first paper that I am aware of to apply it in learning settings (though an argument can be made that Khaw, Louie, Glimcher (2017, PNAS) has elements of a learning task). That it took 15 years of empirical research to explicitly approach the learning setting should be evidence enough of how its proponents were conceptualizing it. The 2011 review paper that the authors interpret as making this claim simply notes that the matching law implies that "the primary determinant of behaviour is the relative value of rewards." (pg 17). In the immediate sentence following they pose the question: "Does the brain represent action values in absolute terms independent of the other available options, or in relative terms?" to set up the remainder of the paper. The use of the term "action values" by Louie and Glimcher is critical. This is neuro-physiological terminology used to describe the motor/decision stage, where the values that have been learned (by the then-standard RL and RPE circuitry covered in pg 14-16) are used to make decisions (these are the Vs in the 2011 paper, or the Qs in the author's model). It is these Vs that are transformed at the time of decision. See also Webb, Louie, Glimcher (pg 4) for a clear statement of this. Nowhere in these articles is it claimed that the action values in learning settings (again, the Vs!) are formed in a relative manner (like in the authors' Equation 4, 5, and 9 for their Qs).

Of course, this point does not diminish the fact that the authors are testing for DN during the updating step in learning, with some compelling evidence. But we need to be clear on where the DN theory has been largely proposed by previous literature, and where the DN theory has been previously shown to capture behaviour (choice sets with more than two options). This is what I mean when I say it is a hypothesis the authors (or their previous reviewers) are proposing and evaluating in their papers. There are perfectly good reasons to examine DN in this setting (see the normative discussion below) and that should be sufficient to frame the question. I believe these issues need to be framed clearly by the introduction of the paper, rather than as an aside at the end of the Discussion.

line 61: This seems like an ideal spot to clearly state that nearly all empirical applications of DN apply it at the decision stage and address choices with more than two options. In particular, the study by Daviet and Webb implements a protocol similar to the design used here but in a "descriptive" rather than a learning setting (see comment below). The text could then note that "Few recent studies have extended…."

As a final point, if the design included three or more alternatives in their transfer phase, there is some reason to believe that behaviour would depart from the model the authors propose (i.e. we might start to see behaviour influenced by more options at the decision stage). Therefore, contrary to the author's claim, it would "matter if they are tested in a binary way". Therefore the results should only be interpreted as applying to the learning of the valuations, not their use during decision. I agree with the changes to lines 141-145, however, the paper up to this point has still not stated that DN at the decision stage has no strong predictions in binary choice. The reader thus has no context on why that statement of line 141 is true. This needs to be clarified.

2) I am re-stating here my comment from Round 1: Previous studies add alternatives to a choice set and study value normalization. They are important distinctions from this work because they test for choice set size effects explicitly. Exp 2 in Webb et al. 2020 does such an experiment (albeit not in a learning setting) and finds support for DN rather than RN. The key difference is that the additional alternatives are added to the choice set, rather than only in a learning phase. Daviet and Webb also report a double decoy experiment of description-based choice, which follows a similar intuition to this experiment by placing an alternative "inside" two existing alternatives (though again, directly in the choice set rather than a learning phase). The conclusions from that paper are also in favour of DN rather than RN since this interior alternative appears to alter choice behaviour. This latter paper is particularly interesting because it relies on the same intuition for experimental design as in the experiment reported here. The difference is just the learning vs. choice setting, suggesting this distinction is critical for observing predictions consistent with RN vs DN.

R2.5 We thank the Reviewer for pointing another important reference that we originally missed. The suggested paper is now cited in the relevant part of the discussion (which is copied in R2.4 above).

I'm not sure that the authors interpreted my comment correctly. I don't understand citation #64 in the Discussion or what that paper has to do with risk. I raise the Daviet and Webb study because it implemented a similar design intuition as this study (inserting an option between two options so as not to alter the range), except with "described" alternatives and not learned outcomes. The conclusions in this paper differed from those reported. The manuscript should clearly state this similarity (and important difference in conclusion) between the two as context for the paradigm the authors are using.

3) In Discussion:

a) lines 361-371: This paragraph argues that "We find no evidence for such a [choice set size] effect". As noted in my previous report (and again above), this study is not examining choice set size effects. The size of the choice set in the transfer phase is always binary. Instead, the study is examining the relative coding of learning. This paragraph needs to be re-written so that this is clearly stated. Referring to the results from the transfer task as a "trinary decision context" is confusing.

b) "However, it is worth noting that evidence concerning previous reports of divisive normalization in humans has been recently challenged and alternatively accounts, such as range normalization, have not been properly tested in these datasets 29,65."

The test for Range normalization in the original Louie et al. dataset is in Webb, Louie andGlimcher 2020, Mng Sci and is rejected in favour of DN. It is not clear what the authors mean by "properly."

4) R2.3b In the ex-ante simulations, we randomly sampled both of these parameters from Β(1.1,1.1) distributions, which inherently leads to different predictions in the transfer phase for the binary and trinary contexts, because the choice rate impacts the update of each option. The fact that small differences derive from allowing different learning rates for obtained and forgone outcomes, is confirmed by simulations. We nonetheless kept the original simulations, because they correspond better to the models that we simulated (with two learning rates). The authors are arguing that there is a systematic difference in the predictions for Range WT_86 WB 86 when using two learning rates. This is not an obvious result, but the increase to 500 simulations seems to confirm that it is systematic. This should be clarified in the text. Otherwise, it will be confusing to readers why the predictions change.

5) From Round 1: line 415 – what is meant by "computational advantage" here? The rational inattention literature, which provides the normative foundation of efficient encoding, places emphasis on the frequency of stimuli (which is completely ignored by range normalization) (e.g. Heng, Woodford, Polania, *eLife*).

Reply: "R2.18 Range and Divisive normalization both provide, to some extent, the computational advantage of being a form of adaptive coding. If the topic interests the Reviewer, we could point to a recent study where their normative status is discussed: https://papers.ssrn.com/sol3/papers.cfm?abstract_id=4320894"

The normative argument for relative coding (including both DN and RN) that the authors point to is specific to a learning context (via the exploration/exploitation tradeoff). And even then, it is not an argument that an efficient code should use only the range. Thus it is important to keep in mind that the results reported in the current manuscript may be somewhat specific to the learning setting implemented in the experimental design. Multiple theoretical and empirical papers from across neuroscience and economics have examined the form of normative computation in general settings (Heng et al., cited above; Steverson et al. cited in previous report; Netzer 2009 AER; Robson and Whitehead, Schaffner et al., 2023, NHB) and all have shown that the efficient code adapts to the full distribution of stimuli. It is reasonable to imagine that some physical systems might approximate this with a piecewise linear algorithm based on the range, but this has not been observed in the nervous system (V1, A1, LIP etc.). Of course this can't rule out the possibility that V1, A1 and LIP get it right with a standard cortical computation, but for some reason other areas have to resort to a cruder approximation. But that isn't the obvious hypothesis. Tracking the minimum and maximum – including identifying which is which – seems both unnecessary and complicated in real-world scenarios. Whether (and how much) the results reported in the paper generalize outside this experimental design is an open question.

---

## [Author Response]

Essential revisions:As you can see in the individual comments there was some confusion about key aspects of your experimental approach and measures. This confusion was resolved during a discussion among reviewers. It would be important that your revised manuscript is more clear about these key aspects. In addition, it would be important to clearly discuss the differences between the predictions made by the different models in choice and learning contexts.

We thank the Editor and the Reviewers for the opportunity to revise and re-submit our manuscript, which we believe has been substantially improved with the help of the Reviewers’ feedback. We were delighted to see that all the Reviewers were positive about our work and we were equally happy to integrate their insightful and constructive comments. In the revised manuscript, we addressed the concerns about the readability of the paper, more specifically the detailed task description in the introduction and Results sections. Moreover, we now include results from two additional experiments designed to better pinpoint the attentional mechanisms behind the weighted normalization process (and exclude choice repetition as a possible interpretation of our results, see Reviewer 3, point 1). All of the other comments have been addressed point-by-point below.

Reviewer #1 (Recommendations for the authors):1. It would be important to more clearly (and early on) state how 'choice rate' was computed for the transfer phase. This may require describing the transfer phase in more detail.

We thank the Reviewer for the helpful suggestion. In line with Reviewer 2’s comments, we improved the Results section by rewriting the transfer explanation in more details:

“For a given option, the transfer phase choice rate was calculated by dividing the number of times an option is chosen by the number of times the option is presented. In the transfer phase, the 10 cues from the learning phase were presented in all possible binary combinations (45, not including pairs formed by the same cue). Each pair of cues was presented four times, leading to a total of 180 trials. Since a given comparison counts for the calculation of the transfer phase choice rate of both involved options, this implies that this variable will not sum to one. Nonetheless, the relative ranking between transfer choice rate can be taken as a behavioral proxy of subjective values.”

2. In lines 272 and 299, please don't use 'perfectly' to qualify the match between model predictions and behavior. I agree that the match is pretty close, but perfectly implies an errorless match, which is not the case here. The authors could consider using 'closely captures' and 'closely match' instead.

This is a fair point, we replaced “perfectly” with “closely” in both sentences, also according to Reviewer 2’s suggestions (Reviewer 2, point 1).

3. There are several typographical errors throughout the manuscript. I am listing a few below but I'd encourage the authors to proofread the manuscript.

Thanks for all, we corrected all these (and other) typos found in the submitted manuscript.

– Line 190: "WB_86 and WT_86", not "NB_86 and NT_86".

Thank you, we’ve made the correction.

– Line 367-368: parenthesis is not closed.

Thank you, we’ve made the correction.

– Line 377-378: parenthesis is not closed.

Thank you, we’ve made the correction.

– Line 382: "that value representations are" not "that values representations are".

Thank you, we’ve made the correction.

Reviewer #2 (Recommendations for the authors):1) The main issues with the paper are expositional. Both a clarification of the simulations used to generate the predictions (Figure 1C) and a clarification of the predictions and appropriate setting of each model in previous literature is needed. In particular, DN makes no relevant predictions in a binary choice setting, so in my initial ignorance, I simply assumed the transfer phase was a trinary choice (because otherwise, DM has no bite). I was so confused by these issues that I needed a discussion with the editor after I submitted my initial report. I have left these comments in my report below (points 1-3 below) so the authors are aware of this miscommunication and that it might help them improve Figure 1 and lines 110 – 130. Finally, the statement of the results (and discussion) is rather opinionated. The writing could do with more nuance regarding the predictions of each model, the appropriate setting for those predictions, and the patterns in the data. In particular, there appear to be patterns that also reject a range normalization account, suggesting that we still have much to learn on this topic.

Concerning the first point, we thank the Reviewer for pointing out the lack of clarity on a crucial aspect of the design (specifically, the relationship between the learning phase and the transfer phase). In the revised manuscript, we better explain the rational between the learning phase (binary vs. trinary choice with feedback) versus the transfer phase (binary choice without feedback).

In our task, because feedback is absent in the transfer phase, values are only formed (or acquired) during the learning phase. The outcome presentation stage is precisely where (range or divisive) normalization is supposed to happen and affect the values of the options (see also Louie et al., 2022). Once values have been acquired during the learning phase (and affected by a normalization process that may or may not depend on the number of outcomes), it does not matter if they are tested in a binary way. We better explained this rationale in the current version of the manuscript:

“Crucially, even if the transfer phase involves only binary choices, it can still tease apart the normalization rules affecting outcome valuation during the learning phase. This is because transfer choices are made based on the memory of values acquired during the learning phase, where we purposely manipulated the number of options and their ranges of values, in order to create learning contexts that allow to confidently discriminate between the two normalization accounts.”

Concerning the second point, we acknowledge that the use of the word “perfectly” may not have be warranted in our case (or probably is not in general in empirical sciences) and, accordingly, we removed its occurrences in favor or more modest nuanced terms (such as “closely”). We also better and sooner highlight the fact that, even though divisive normalization has been proposed in the context of reinforcement learning in the way we implement it, it has been mainly used to explain description-based choices do represent a difference compared to our design:

“It should be noted here that, although divisive normalization has been more frequently applied to the prospective evaluation of described outcomes (e.g., lotteries; snack-food items), rather than retrospective evaluation of obtained outcomes (e.g., bandits), it has both historical^21^ and recent^25^ antecedents in the context of reinforcement learning.”

Before discussion with the editor:2) The choice probabilities do not sum to 1 within condition (for example, in treatment NB, the "50" option seems to be chosen with probability.66 and the "14" with probability <.2. Similar issues arise for DIVISIVE and RANGE. Either I deeply misunderstand the experimental setting, or there is something wrong with the simulation underlying the figure (the simulation uses an argmax, so I do not see how the probabilities can not sum to 1).3) Why are the unbiased choice probabilities for "50" equal across NB and NT in UNBIASED? Loosely speaking, the Q-learning rule used by the authors will generate a Q_45 that is normally distributed around means [14,50,32,86] (see SuttonandBarto Equation 2.6 and impose a CentralLimitTheorem). Taking the arg-max of these normal random variables will require that P(50) (weakly) decreases as the set size increases. The authors seem to be claiming that it is constant.4) In an effort to find the causes of these differences, I attempted to simulate the model. Apart from a number of issues with the description of the computational model (see below), the reported simulation size leads to substantial variation between simulation runs (with different RNG seeds).– Note that I did not include heterogeneity in the learning rate in these simulations, since the theoretical setting does not require it (at least the authors don't claim that it does). So α is set to 0.5, which is the mean of the Β(1.1,1.1) distribution the authors use.

We thank the Reviewer for raising these points, and for even taking the time to make simulations. Since these points have been reformulated after discussion with the editor, we refer to the points below for detailed responses, that, we hope address the questions raised by the Reviewer (while contributing improving the manuscript).

After discussion with the editor:5) Line 123: "It is calculated across all the comparisons involving a given option." This seemingly innocuous sentence needs much more explanation. Does the transfer phase include ALL binary choices between ALL stimuli, including those across conditions? So is the "86" from the WB paired with the "86" from "WT"? If so, presumably this is what is generating the identical choice rates in UNBIASED since they are essentially the same Q_45(86). It is still deeply unsettling that nothing adds up to 1 in Figure 1C, but at least now I understand the source of the differing predictions. However, there still are some issues:

We thank the Reviewer for the opportunity to clarify. Indeed, the transfer phase involves binary choices between all possible combinations (among 10 stimuli, hence 45 combinations) that were repeated 4 times in order to increase the reliability. For a given option, we indeed plot the number of times it has been chosen, considering all comparisons involving the given option (and this is why the rates do not sum up to 1). We understand that this may be confusing if one is not familiar with this paradigm and this is why we revised the manuscript (Results and Figure 1s’ legend) in order to make it clearer:

Results section

“For a given option, the transfer phase choice rate was calculated by dividing the number of times an option is chosen by the number of times the option is presented. In the transfer phase, the 10 cues from the learning phase were presented in all possible binary combinations (45, not including pairs formed by the same cue). Each pair of cues was presented four times, leading to a total of 180 trials. Since a given comparison counts for the calculation of the transfer phase choice rate of both involved options, this implies that this variable will not sum to one. Nonetheless, the relative ranking between transfer choice rate can be taken as a behavioral proxy of subjective values.”

Legend of Figure 1

“Note that choice rate in the transfer phase is calculated across all possible binary combinations involving a given option. While score is proportional to the agent’s preference for a given option, it does not sum to one, because any given choice counts for the final score of two options.”

The reviewer may also find useful Figure 3 —figure supplement 2 that has been used by our and other groups (Bavard et al., 2018; Hayes et al., 2022), where each individual transfer phase choice is presented:

– Why is there a slight increase in the choice proportion for 86 between WT and WB? Shouldn't Q_45 be the same for them because the range doesn't change?

This is an interesting point. In our simulations, as in our optimization procedures, we allowed the models to learn differently from the chosen and the unchosen option(s). In other words, as in previous studies (e.g., Palminteri et al., 2015), all models include two different learning rates: factual (chosen option) and counterfactual (unchosen option(s)). This is implemented to account for the (reasonable) possibility that less attention is given to the unchosen option(s).

In the ex-ante simulations, we randomly sampled both of these parameters from Β(1.1,1.1) distributions, which inherently leads to different predictions in the transfer phase for the binary and trinary contexts, because the choice rate impacts the update of each option. The fact that small differences derive from allowing different learning rates for obtained and forgone outcomes, is confirmed by simulations. When using only one learning rate for all option updates, the slight increase in the choice proportion disappears:

We nonetheless kept the original simulations, because they correspond better to the models that we simulated (with two learning rates).

– The number of simulations should be increased (i.e. is it just "10 choose 2" x 50 choices). Given comment 3 above, are we sure there is no noise in the predictions (see points above)? Also, the issue of introducing α heterogeneity into the simulation needs to be addressed, as it seems pointless.

Since the original version of the experiment included a sample size of 50 participants, we chose to run model predictions with the same number of simulations. However, we agree that we need to make sure that increasing the number of simulations will not change the predictions: we therefore simulated the same models but increased the number of simulations from 50 to 500. The results suggest that increasing the number of simulations does not change the model predictions:

**Author response image 1. sa2fig1:** Model predictions generated from 500 simulations.

Since the simulation results are identical, we prefer to keep the 50 run simulations in order to keep coherent with our actual sample size (and to minimize changes in an otherwise quite abundantly revised manuscripts).

6) The framing in the introduction of the paper needs to be clear about the use of a binary choice transfer phase. DN, as conceptualized by previous work the authors cite, is a model for how valuations are normalized in the choice process (e.g. Webb et al., Mng Sci, 2020, Figure A.4). As such, it is acknowledged that it has limited prediction in binary choice (see Webb et al., Mng Sci, 2020 Proposition 1). Nearly all of its testable implications arise in trinary (or larger) choice sets. For example, the quote in Footnote 1 is the first sentence of the first paragraph of a section titled "Choice Set Size Effects" which describes studies in which there are more than two alternatives in the choice set. There is no choice set size effects in this experiment because it only examines binary choice in the transfer phase, therefore it excludes the types of choice sets in which DN has been previously observed.As such, the authors write that they opt for a reinforcement learning paradigm "because it has greater potential for translational and cross-species research". It may indeed have translational benefits for cross-species research, and learning settings are important in their own right, but the use of an RL paradigm is not innocuous. There may be critical differences in the form of value coding computations between conditioned learning settings and "descriptive" settings in which valuations are already known (as acknowledged by the authors in an earlier ScienceAdvances paper), not the least of which might be due to the differences between cortical and sub-cortical neural computations (the neural evidence for DN is primarily in cortical regions). There is only one paper I am aware of that applies DN in an RL setting (Louie, PLOSCompBio2022) to account for gain/loss asymmetry in behaviour without the need for different learning rates (though not cited by the authors). This paper should make clear that the application of DN to a conditioned learning phase is a hypothesis raised in this paper, and clarify the distinction between settings in which DN has been previously proposed and examined in other work.

The Reviewer here raises a good point (we note that a similar point was also raised and, we hope, addressed in Reviewer2, poinry1, even if we happy to provide additional, related, arguments below). In short, we agree with the Reviewers that our introduction needed to better articulated, including the transition between the current literature on divisive normalization and our study. And we also agree that Louie (2022)’s paper should be mentioned in the introduction, since it nicely set the stage concerning how divisive normalization should be logically implemented in the reinforcement learning framework. This is how we modified the introduction:

“Even though, to date, most of the empirical studies proposing divisive normalization as a valid model of economic value encoding proposed that option values are vehiculated by explicit features of the stimulus (such as food snacks or lotteries: so-called described options^22,17,23^), few recent studies have extended the framework to account for subjective valuation in the reinforcement learning (or experience-based) context^24,25^. Adjusting the divisive normalization model to a reinforcement learning scenario is easily achieved by assuming that the normalization step occurs at the outcome stage, i.e., when the participant is presented with the obtained (and forgone) outcomes.”

Concerning the discussion, we expanded the paragraph where we discuss the fact that things may be different in description-based choices (and included more citations):

“Multiple elements of our results concordantly show that divisive normalization does not provide a good account of subjective value representation in human reinforcement learning. More precisely, we were concerned about whether at the outcome stages, the subjective values of rewards were normalized according to a divisive (or range) normalization rule^24,25^. It is nonetheless still possible that this rule provides a good description of human behavior in other value-based decision-making domains. In fact, most of the previous studies claiming evidence for divisive normalization used other tasks involving items whose values are described (such as snacks or lotteries food) and have not to be extracted from experience^17,19,26,63^. In addition, our study only addressed value normalization only the outcome magnitude space and we did not address whether the same rule would apply to other outcome attributes, such as probability^64^.”

7) Previous studies add alternatives to a choice set and study value normalization. They are important distinctions from this work because they test for choice set size effects explicitly. Exp 2 in Webb et al. 2020 does such an experiment (albeit not in a learning setting) and finds support for DN rather than RN. The key difference is that the additional alternatives are added to the choice set, rather than only in a learning phase. Daviet and Webb also report a double decoy experiment of description-based choice, which follows a similar intuition to this experiment by placing an alternative "inside" two existing alternatives (though again, directly in the choice set rather than a learning phase). The conclusions from that paper are also in favour of DN rather than RN since this interior alternative appears to alter choice behaviour. This latter paper is particularly interesting because it relies on the same intuition for experimental design as in the experiment reported here. The difference is just the learning vs. choice setting, suggesting this distinction is critical for observing predictions consistent with RN vs DN.

We thank the Reviewer for pointing out another important reference that we originally missed. The suggested paper is now cited in the relevant part of the discussion (which is copied in Reviewer 2, point 4 above).

8) The experiment tests RN vs DN by expanding the range upward with the "86" option. However, RN would make an equivalent set of predictions if the range were equivalently adjusted downward instead (for example by adding a "68" option to "50" and "86", and then comparing to WB and WT, because it effectively sets the value to "0" or "1" depending only on the range). The predictions of DN would differ however because adding a low-value alternative to the normalization would not change it much. Given the design of the experiment, this would seem like a fairly straightforward prediction to test for RN. Would the behaviour of subjects be symmetric for equivalent ranges, as RN predicts? If so this would be a compelling result, because symmetry is a strong theoretical assumption.

We addressed this point above (see Reviewer 2, point 1), but, in short, we ran three versions of the suggested experiment (N=50 each) and the results confirm our conclusions.

9) There were issues with the reporting of the results that need to be addressed:line 183, typo NB_50?

Thank you, we’ve made the correction.

lines 182-188 need copy-editing.

Thank you, we’ve made the corrections.

line 186; "numerically lower" has no meaning in a statistical comparison. The standard error (t-test) is telling us that there is too much uncertainty to determine if there is a difference between treatments. No more, no less. Arguing for "numerical" significance is equivalent to arguing that there is signal in noise. Statements like this should be removed from the paper.

Thank you, we agree with the Reviewer and deleted this sentence from the paper.

line 188 – "superficially"? a test either rejects a prediction or doesn't. More specifically, this observation also rejects a prediction of RN and should be stated as such for clarity.

Correct. We modified the sentence accordingly:

“Concerning other features of the transfer phase performance, some comparisons were consistent with the UNBIASED model and not with the RANGE model, such as the fact that high value options in the narrow contexts (NB_50_ and NT_50_) displayed a lower choice rate compared to the high value options of the wide contexts (WB_86_ and WT_86_; t(49)=-4.19, p=.00011, d=-0.72), even if the size of the difference appeared to be much smaller to that expected from ex ante model simulations (Figure 1C, right).”

line 183-184 – given that experiments a, b, and c, seem similar and are pooled in later analyses, it would be useful to also pool them to test whether there is an increase in the NT_86 vs NB_86 and WT_86 vs WB_86 choice rates (perhaps using a pooled paired t-test). A tripling of the number of observations might be helpful here for testing whether NB_50<nt_50<wb_86.

This is an interesting point that we indeed did not address in the previous version of the manuscript. To our understanding, this can be addressed by making two comparisons:

1) We pooled most favorable options from the narrow (NB50 + NT50) contexts on one side, and the wide (WB86 + WT86) contexts on the other side, and we compared them: t(149)=-7.42, p<.0001, d=-0.61. This effect, already mentioned in the Results section (see previous point) can be explained by the fact that some participants are best explained by the UNBIASED or the DIVISIVE models, which both account for this effect. Another explanation, in line with our previous results suggesting that context-dependence in reinforcement learning arises progressively over the task time (Bavard et al., 2018, 2021), could be that values are encoded on an objective (absolute) scale at the beginning of the learning experiment, and are progressively rescaled into context-dependent (relative) values. As we chose not to include such parameters in our models, none of them could capture these potential dynamic effects.

2) We pooled the most favorable options from the binary (NB50 + WB86) contexts on one side, and the trinary (NT50 + WT86) contexts on the other side, and we compared them: t(149)=-4.11, p<.0001, d=-0.34. This result actually provides *strong* evidence against the DN account, which would predict a significant effect in the *opposite direction*. However, it is true that this effect is not predicted by the RN account either. We believe that this effect (i.e., over-evaluating the most favorable options in trinary contexts compared to the binary ones) may arise from additional (not modelled) valuation processes. A plausible process could be to suppose that the most favorable options in trinary contexts receive a bonus because they have been previously identified as being better than two options (rather than just one as in the binary contexts). This interpretation is consistent with instance-based (or more indirectly with decision-sampling) accounts of decision-making. This observation is interesting and will motivate further research in our (and hopefully also other) teams, involving decision contexts with more than 4 options, but it does not affect our main claim (strong falsification of the DN account). We mentioned this result in the revised manuscript:

Results:

“One feature was not explained by any of the models, such as the higher choice rate for the high value options in the trinary contexts (NT_50_ and WT_86_) compared to the binary contexts (NB_50_ and WB_86_; t(49)=3.53, p=.00090, d=0.50) Please note that, while the statistical test is significant in Experiment 1a, it stays so when taking into account all experiments (t(149)=4.11, p<.0001, d=0.34). Of note, the direction of the effect for this comparison is in stronger contrast with the DIVISIVE (which predicts a difference in the opposite direction) rather than the RANGE and UNBIASED models (which predict no difference).”

Discussion:

“Despite the fact that including the (attentional) weighting parameter improved the quality of fit (both in terms of out-of-sample log-likelihood and model simulations) of range normalization process, we acknowledge that some features of the data were still not perfectly accounted. For instance, even if the effect was small in size, from averaging across several experiments it appeared that choice rate for the high value options in the trinary contexts were higher compared to those in the binary ones. Although this feature provides strong evidence against the divisive normalization framework (which predicts the opposite effect), it is also not coherent with the range normalization process. It could be hypothesized that other cognitive and valuation mechanisms concur to generate this effect, such as instance-based or comparison-based decision valuation processes where the options in the trinary contexts would benefit from an additional (positive) comparison^68,69^.”

line 272 – "perfectly" is an overstatement, given the log-likelihood is not zero. Also see the NB_50<nt_50<wb_86

We agree with the Reviewer, and replaced “perfectly” with “closely”, also following Reviewer 1’s suggestion.

line 299 – perhaps also not "perfect" What is the t-test reported here? And how to interpret this p-value of.13? Is this an argument for accepting the null hypothesis?

We agree with the Reviewer, and replaced “perfectly” with “closely”, also following Reviewer 1’s suggestion. We thank the Reviewer for pointing out that a t-test might not be the most appropriate way of comparing simulations (using the explicit values) and behavioral data in this case. To better account for the comparison, we reported in the revised text the correlation between simulated and behavioral data per option (Experiment 1: Spearman’s *ρ*(8)=0.99, *p*<.0001; Experiment 2: Spearman’s *ρ*(8)=0.99, *p*<.0001; Experiment 3: Spearman’s *ρ*(10)=1.00, *p*<.0001):

**Author response image 2. sa2fig2:** Simulated and behavioral data for the Explicit phase. Top: behavioral choice-based data (black dots) superimposed on simulated choice-based data (colored bars). Simulated data were obtained using the explicit ratings as values and a argmax decision rule (see Methods). Bottom: average simulated data per option (horizontal error bars) as a function of average behavioral data per option (vertical error bars). Error bars represent s.e.m., plain diagonal line represents idendity.

Reviewer #3 (Recommendations for the authors):I have a few comments for the authors to consider:1) Training performance (Figure 2) seems very close to the ceiling (i.e. accuracy of 100%) for many of the participants. I would be slightly worried that there isn't enough stochasticity for some participants to derive meaningful parameters from the model – did the authors worry about this or run any checks to guard against this possibility at all? Perhaps more interestingly, this pattern (which suggests participants quickly identify the best option and just keep selecting this) also made me wonder whether the unbiased model with a persistence parameter (i.e. which bonuses the most recent choice) would fare as well in terms of log-likelihood scores (Table 2) and/or capture the same pattern of probe (transfer phase) choices as either the divisive or range models. Could the authors refute this possibility?

We thank Prof. Garrett for this interesting point that we separate in two parts. The first one, concerning whether or not the models are recoverable in our task (even assuming the high level of performance observed). We checked and this is the case (see Author response image 3). This being said, we note that the specific pattern we are interested in (namely post-learning transfer phase choices) it is quite robust to the choice of the parameter (we ensured that by running ex ante model simulations using specifically distributed parameters (Daw et al., Neuron 2011)) and our claims are based on transfer phase choice.

**Author response image 3. sa2fig3:** Confusion matrix for model recovery. For each model, we performed ex-ante simulations following the procedure described in the Methods section. We then fitted the simulated data with each model and computed the out-of-sample log-likelihood on the transfer phase data. Each cell depicts the frequency with which each model (columns) is best predictive for the simulated data for each model (rows).

The second point concerns possible confounds that could arise, assuming that the participants might present a form of choice repetition which is not included in the model. Indeed, we were well aware of this possibility and this is why in our first publication about the topic we included (and rejected) a model (that we called HABIT) that precisely instantiated this idea. We refer to Bavard et al. (2021) for more information about the results (see also Palminteri, Behavioral Neuroscience 2023, for a, perhaps even deeper, discussion of the perseveration model in the context of optimistic update).

Of course, this does not automatically guarantee that in our current set-up, choice repetition or perseveration do not play a role. This is why, following Prof. Garrett’s suggestion, we tested the habit model and show that, while it quantitatively (yet not so much qualitatively) outperforms the UNBIASED model, it does not work as well as the RANGE^w^ model (oosLL_HAB_ vs oosLL_UNB_ : *t*(149)=9.55, *p*<.0001, *d*=0.26; oosLL_HAB_ vs oosLL_RAN_^w^ : *t*(149)=-7.16, *p*<.0001, *d*=-0.72; see Author response image 4).

**Author response image 4. sa2fig4:** Model predictions of the UNBIASED (left), RANGE^w^ (middle) and HABIT (right). Simulated data in the transfer phase were obtained with the best-fitting parameters, optimized on all four contexts of the learning phase.

Finally, in addition to running the supplementary analyses reported here (and because we thought that these experiments could help better address the cognitive/attentional mechanisms value normalization), we ran two additional experiments in some specific choice contexts, where we prevented the participant to choose the maximum value option. We therefore managed to orthogonalize *option values* from *option choosiness*, because in the conditions where the maximum option could not be chosen, “option choosiness” is higher for the mid-value option. The key idea behind this manipulation is that, in this design, if preferences in the transfer phase derive from a choice repetition mechanism, the mid options should be preferred to the max options (of the forced choice contexts) and the max option of the free choice contexts should be preferred to the max option of the free choice contexts. In short, neither of these two predictions of the choice repetition model are fulfilled by the experiment (accordingly the out-of-sample log-likelihood of the choice repetition model, which is much lower compared to that of our winning model). These experiments, whose main results, methods and figures are reported below for convenience, do inform us however about the attentional mechanisms underlying value normalization.

Results:

“Investigating the attentional mechanisms underlying weighted normalization

However, our design, as implemented so far, does not allow to tease apart two possible mechanisms underlying subjective weighting of outcome captured by power transformation. One possibility (implicit in the formulation we used) is that participants “perceive” mid outcomes as being closer to the low one, because the high outcome “stands out” due to its value. Another possibility is that participants give a higher subjective weighting to chosen outcomes, because of the very fact that they were chosen and obtained. The current design and results do not allow to tease apart these interpretations, because during the learning phase the mid values options were chosen as much as the low value options (7.2% and 6.8%, t(149)=0.97, p=.33, d=0.04) and therefore mid outcomes were almost systematically unchosen outcomes.

To address this issue, we ran two additional experiments (Experiment 3a and 3b), featuring, as before, wide and narrow learning contexts (Figure 5A). The key manipulation in this new experiment consisted in leaning contexts where we interleaved trinary choices with binary choices, where the high value option was presented, but not available to the participant (Figure 5B). We reasoned that by doing so, we would be able to increase the number of times the mid value options were chosen. The manipulation was successful in doing so: in the learning contexts featuring binary choices, the mid value options were chosen on 48% of the trials (Experiment 3a) and 67% (Experiment 3b); significantly more than the corresponding high value option in the same learning context (Experiment 3a, wide: t(99)=6.03, p<.0001, d=0.95; narrow t(99)=5.43, p<.0001, d=0.80; Experiment 3b, wide: t(99)=33.27, p<.0001, d=4.47; narrow t(99)=34.06, p<.0001, d=4.33; Supplementary Figure S7).

We then turned to the analysis of the transfer choices and found that the manipulation was also effective in manipulating the mid option value, so that in the contexts featuring binary choices (i.e., impossibility of choosing the high value options), the mid options were valued more compared to the full trinary contexts (i.e., when they were almost never chosen) (Experiment 3a, wide: t(99)=22.80, p<.0001, d=3.46; narrow: t(99)=20.10, d=3.06, p<.0001; Experiment 3b, wide: t(99)=21.96, p<.0001, d=3.88; narrow t(99)=20.46, p<.0001, d=3.76; Figure 5C). Interestingly, the results were virtually identical in the experiment with 50% and that with 25% trinary trials, despite the choosiness of the high value options being very different in the two experiments and the signatures of range adaptation (narrow vs. wide) were replicated (and we therefore pooled the experiment in the main figure).

The behavioral results thus suggest that mid outcomes, although range normalized, can be valued correctly in between the lowest and the highest outcome, if we force choices toward the mid value option. These results are therefore consistent with the hypothesis that outcome weighting is contingent with option choosiness and a bias in outcome evaluation per so. To objectify this conclusion, we compared the RANGE^w^ previously described, with a more complex one (RANGE^w+^) where two different power ω parameters apply to the obtained (chosen: ωc) and forgone (unchosen: ωu) outcomes. This augmented model displayed better higher quality of fit in both experiments (as proxied by the out-of-sample log-likelihood of the transfer phase; oosLL_RAN(ω)_ vs oosLL_RAN(ω+)_ : t(199)=-7.73, p<.0001, d=-0.30). This quantitative result was backed up by model simulations analysis showing that only the RANGE^w+^ was able to capture the change in valuation in the mid value options (Figure 5C). Finally, we compared the weighting parameters and found ωc significantly lower then ωu (t(199)=-17.28, p<.0001, d=-1.92; Figure 5D). To conclude, these additional experiments further clarify the cognitive mechanisms (and specifically the role of attention) underlying outcome encoding.”

Discussion:

“We believe that this may derive from attentional mechanisms that bias evidence accumulation as a function of outcomes and option expected values^5,51,52^. To further probe this hypothesis, we designed and ran an additional experiment (Experiment 3) where we manipulate the possibility of choosing the high value option in trinary learning contexts. This manipulation successfully managed to “correct” the subjective valuation of mid-value options (while leaving unaffected the valuation of the other options). The behavior in this experiment was successfully captured by further tweaking the weighted range normalization model by assuming that different weighting parameters applies to chosen and unchosen outcomes. By finding concave and convex weighting functions for chosen and unchosen outcomes, respectively, the model managed to explain why forcing the participant to choose the mid value option increases its subjective valuation. We believe that these results further reinforce the hypothesis that outcome valuation interacts with attentional deployment. In fact, it is reasonable to assume that the obtained outcome is attended more than the forgone ones (after all it is the more behaviorally relevant outcome) and that increased attention devoted to the obtained outcomes “boosts” its value^51^. This effect can also be conceptually linked to a form of choice-confirmation bias, where the mid value outcome is perceived as better than it actually is^42^.”

2) Previous models from this group (e.g., Palminteri 2015 Nature Comms) have conceptualised "contextual value" as an iteratively learned quantity (like the Q values here). But in the normalisation models presented, this aspect is absent. For example, min(R) and max(R) in equation 8 used for the range normalisation model are simply the min/max rewards on that trial (or on the previous trial when full feedback is absent). Similarly, the denominator in equation 9 for the divisive normalisation model. In practice, these could each be updated. For example, instead of min(Ri) in eq. 9, they could use something along the lines of Vmin(i) which updates as:Vmin(i)t+1 = Vmin(i)t + α*δ tδt = min(Ri) – Vmin(i)tDid the authors consider this? Or, another way of asking this, is to ask why the authors assume these key components of the model are not learned quantities that get updated over trial-by-trial experience.

Prof. Garrett correctly identifies the fact that our normalization variable in the current study are not latent or inferred, but rather extracted from outcome information on a trial-by-trial basis. This is indeed different from most of our previous models (see Palminteri and Lebreton, COBS, 2021). One reason we opted for this formulation is that in the current design, it is quite *plausible*. First, feedback is always complete, so the participant gets at every trial a fairly good picture of the range. This is not the case of course in partial feedback scenarios (half the experiments of Bavard et al., 2018; 2021). In addition to being complete, the feedback was – although stochastic with Gaussian noise – continuous and not binary (or Bernoullian). For instance, even in the complete feedback experiments of Bavard et al. (2021), it was quite possible that in a given trial the subjects show something like [Rc=0; Ru=0] or [Rc=10; Ru=10] and one case see why in this scenario it may be sound to not rely on min(R) or max(R), but rather keep more stable (latent) variable. The second reason is that the current version is more *parsimonious*. In fact, once showed that it works in our set-up, why bother with an additional free parameter? But we of course recognize that shall we move to a partial feedback scenario, latent estimates of Rmax(s) and Rmin(s) will become necessary.

We acknowledge this fact in the revised discussion:

“Finally, further experiments will be needed to generalize the current models to partial feedback situations where the contextual variables have to be inferred and stored in memory.”

3) Transfer phase – the transfer phase only contains binary combinations. However, during training participants were presented with two binary combination choices and two trinary combination choices (i.e. choices were 50% binary choices and 50% trinary). Could exclusively using binary combinations in the probe phase bias the results at all towards finding stronger evidence for the range model? I'm thinking (and I could be wrong!) that the divisive approach might be better suited to a world where choices are over >2 options. So perhaps participants make choices more consistent with the range model when the probe phase consists of binary choices but might switch to making choices more consistent with the divisive model if this consisted of trinary choices instead. The explicit rating (Figure 5) results do go some way to showing that the results are not just the result of the binary nature of the probe phase (as here ratings are provided one by one I believe). Nonetheless, I'd be interested if the authors had considered this at all or considered this a limitation of the design used. </nt_50<wb_86</nt_50<wb_86

Prof. Garrett here asks whether “*using* binary combinations in the probe phase bias the results at all towards finding stronger evidence for the range model?” The short answer of this question is “no”. To understand why, one has to first consider that in our task, outcomes are received only during the learning phase. During the learning phase, outcomes could be ternary of binary and here is the key step/phase where the two models diverge in how value is processed. The second point to consider is that the simulations, ex ante and ex post alike show that the three models are perfectly discriminable, so as far as one accepts that we are correctly instantiating divisive and range normalization in the context of RL (and we believe Prof. Garrett does so), one should also accept that our set-up allows to discriminate among them without providing unfair advantage to any of them (Please also see some related responses to Reviewer 2).

[Editors' note: further revisions were suggested prior to acceptance, as described below.]

The manuscript has been improved but there are some remaining issues that need to be addressed, as outlined below:As you can see, two of the original reviewers were satisfied with your revisions. However, Reviewer #2 does not feel some of their points were fully addressed. Most importantly, there are open issues regarding the contextualization of the current study as well as the important fact that DN was designed to account for choice set effects, rather than normalization during learning, as tested here. Specifically, the binary choice test is only sensitive to value normalization during learning but not whether the form or normalization of these learned values during choice. In other words, while the current experiments test the interesting and worthwhile question of how values are normalized during learning, they do not test whether DN is still applied to RN-learned values at the time of choice. In summary, there is an important distinction between the design of the current study and what has been done to study DN in the past as well as how DN was originally conceptualized. We agree with Reviewer #2 and feel that this distinction should be acknowledged and clarified throughout the paper, most importantly early on in the introduction. Also, the discussion would benefit from revisions that restrict the conclusions of the findings to the appropriate (i.e., learning) context. We are looking forward to receiving a revised version of your interesting and important manuscript.

We thank the Editors for taking care of our submission, their thoughtful and careful comments, as well as the overall positive appreciation of our paper.

We are happy to submit a further revised version of the manuscript that further stress the difference between previous and current literature, together with a point-by-point response to the interesting points raised by Reviewer 2.

Reviewer #2 (Recommendations for the authors):The authors have addressed some of my comments from the previous round, though a few remain. In particular, there still seems to be some disagreement over how to describe the study and place it in the context of previous literature. I realize that the authors are addressing (what they interpret) to be overly-strong general statements about the form of normalization in decision-making. This is a useful undertaking. But repeating the same mistake of over-generalizing and not placing the results in context will only serve to continue the confusion.

We thank the Reviewer for their time and unquestionable dedication in providing a very thorough review. We are also glad to see that, on the important matters (design, analyses and results), we seem to be on the same page. We appreciate that there remains some degree of disagreement between the perception of the claims that have been made in literature concerning the domain of application of DN, even though we believe that (especially in the Revised manuscript) we made some substantial steps to make our literature more accurate (thanks to the thoughtful suggestions of the Reviewer).

We believe that Author response image 5 correctly exemplifies the current state of affair and how we presented things in our revised manuscript. Based on the previous and the current exchanges, it is also clear that the Reviewer agrees with us. Of note, even though Author response image5 would probably not be appropriated for the main text of a published paper, we are glad that *eLife* policy is to publish the Rebuttal letter and that it could be used to further clarify the evolution of the literature concerning the applications of DN to the value-based framework.

Finally, before diving into the specific responses, we hope that the Reviewer did not miss the point that the very reason we got interested in this research question in the first place, is that we found the DN framework extremely convincing and stimulating. We praise all the authors previously involved this line of research to deliver an impressive set of (testable) theories and findings.

**Author response image 5. sa2fig5:** Value-based decision-making framework (as depicted by Rangel et al.) and where DN normalization has been traditionally (“Valuation”) and recently (“Outcome evaluation”) applied.

1) In their reply, the authors write:"The outcome presentation stage is precisely where (range or divisive) normalization is supposed to happen and affect the values of the options (see also Louie et al., 2022). Once values have been acquired during the learning phase (and affected by a normalization process that may or may not depend on the number of outcomes), it does not matter if they are tested in a binary way."and:"First of all, we note that nowhere in the papers proposing and/or defending the DN hypothesis of value-based choices, we could find an explicit disclaimer and/or limitation indicating the authors believe that it should only apply to prospective valuation of described prospects and not to retrospective valuation of obtained outcomes."I fundamentally disagree on this point. There are various opinions on where normalization is "supposed to happen." Multiple papers have conceptualized DN as a process that occurs during decision and not learning (the citations were provided in the previous round, including some that I feel qualified to interpret). Louie et al., 2022 is the first paper that I am aware of to apply it in learning settings (though an argument can be made that Khaw, Louie, Glimcher (2017, PNAS) has elements of a learning task). That it took 15 years of empirical research to explicitly approach the learning setting should be evidence enough of how its proponents were conceptualizing it. The 2011 review paper that the authors interpret as making this claim simply notes that the matching law implies that "the primary determinant of behaviour is the relative value of rewards." (pg 17). In the immediate sentence following they pose the question: "Does the brain represent action values in absolute terms independent of the other available options, or in relative terms?" to set up the remainder of the paper. The use of the term "action values" by Louie and Glimcher is critical. This is neuro-physiological terminology used to describe the motor/decision stage, where the values that have been learned (by the then-standard RL and RPE circuitry covered in pg 14-16) are used to make decisions (these are the Vs in the 2011 paper, or the Qs in the author's model). It is these Vs that are transformed at the time of decision. See also Webb, Louie, Glimcher (pg 4) for a clear statement of this. Nowhere in these articles is it claimed that the action values in learning settings (again, the Vs!) are formed in a relative manner (like in the authors' Equation 4, 5, and 9 for their Qs).Of course, this point does not diminish the fact that the authors are testing for DN during the updating step in learning, with some compelling evidence. But we need to be clear on where the DN theory has been largely proposed by previous literature, and where the DN theory has been previously shown to capture behaviour (choice sets with more than two options). This is what I mean when I say it is a hypothesis the authors (or their previous reviewers) are proposing and evaluating in their papers. There are perfectly good reasons to examine DN in this setting (see the normative discussion below) and that should be sufficient to frame the question. I believe these issues need to be framed clearly by the introduction of the paper, rather than as an aside at the end of the Discussion.line 61: This seems like an ideal spot to clearly state that nearly all empirical applications of DN apply it at the decision stage and address choices with more than two options. In particular, the study by Daviet and Webb implements a protocol similar to the design used here but in a "descriptive" rather than a learning setting (see comment below). The text could then note that "Few recent studies have extended…."As a final point, if the design included three or more alternatives in their transfer phase, there is some reason to believe that behaviour would depart from the model the authors propose (i.e. we might start to see behaviour influenced by more options at the decision stage). Therefore, contrary to the author's claim, it would "matter if they are tested in a binary way". Therefore the results should only be interpreted as applying to the learning of the valuations, not their use during decision. I agree with the changes to lines 141-145, however, the paper up to this point has still not stated that DN at the decision stage has no strong predictions in binary choice. The reader thus has no context on why that statement of line 141 is true. This needs to be clarified.

We thank the Reviewer for this interesting and scholarly arguments. In link with our previous comment, we want to emphasize that the 15 years necessary to conceptualize and operationalize DN in the context of value-based decision-making have been well-spent. The measure of this success is given by the fact that many other groups (including our own) picked up the same or similar research question, thus provoking a dialectical and constructive debate that will push us a step closer to having a satisfactory descriptive (and yet normatively ground) model of human decision-making.

Nonetheless, concerning the specific points, we are a bit puzzled because we agree with the Reviewer, buy we fail to see how the incriminated paragraph does not vehiculate these ideas. Around line 61ish, we believe it is clearly stated that

(1) most of the existing account of DN are framed within descriptive framework:

“Even though, to date, most of the empirical studies proposing divisive normalization as a valid model of economic value encoding proposed that option values are vehiculated by explicit features of the stimulus (such as food snacks or lotteries: so-called described options^22,17,23^),…”

(2) that only few studies applied it to the experience-based domain:

“…few recent studies have extended the framework to account for subjective valuation in the reinforcement learning (or experience-based) context^24,25^.”

(3) and, finally, what is the main computational modification entailed by the description experience translation.

“Adjusting the divisive normalization model to a reinforcement learning scenario is easily achieved by assuming that the normalization step occurs at the outcome stage, i.e., when the participant is presented with the obtained (and forgone) outcomes.”

Similarly, around line 140ish is also clear that normalization is applied to outcome evaluation:

“Crucially, even if the transfer phase involves only binary choices, it can still tease apart the normalization rules affecting outcome valuation during the learning phase. This is because transfer choices are made based on the memory of values acquired during the learning phase, where we purposely manipulated the number of options and their ranges of values, in order to create learning contexts that allow to confidently discriminate between the two normalization accounts, in the reinforcement learning context.”

(of note, we just added “in the reinforcement learning context” to make our domain of application even clearer).

We defer to the Editors’ guidance to find better formulation, if needed, but we maintain that our chain of reasoning is not misleading (and actually quite on line with the views proposed by the Reviewer).

2) I am re-stating here my comment from Round 1: Previous studies add alternatives to a choice set and study value normalization. They are important distinctions from this work because they test for choice set size effects explicitly. Exp 2 in Webb et al. 2020 does such an experiment (albeit not in a learning setting) and finds support for DN rather than RN. The key difference is that the additional alternatives are added to the choice set, rather than only in a learning phase. Daviet and Webb also report a double decoy experiment of description-based choice, which follows a similar intuition to this experiment by placing an alternative "inside" two existing alternatives (though again, directly in the choice set rather than a learning phase). The conclusions from that paper are also in favour of DN rather than RN since this interior alternative appears to alter choice behaviour. This latter paper is particularly interesting because it relies on the same intuition for experimental design as in the experiment reported here. The difference is just the learning vs. choice setting, suggesting this distinction is critical for observing predictions consistent with RN vs DN.R2.5 We thank the Reviewer for pointing another important reference that we originally missed. The suggested paper is now cited in the relevant part of the discussion (which is copied in R2.4 above).I'm not sure that the authors interpreted my comment correctly. I don't understand citation #64 in the Discussion or what that paper has to do with risk. I raise the Daviet and Webb study because it implemented a similar design intuition as this study (inserting an option between two options so as not to alter the range), except with "described" alternatives and not learned outcomes. The conclusions in this paper differed from those reported. The manuscript should clearly state this similarity (and important difference in conclusion) between the two as context for the paradigm the authors are using.

We thank the Reviewer for clarifying this point. For the sake of completeness (even if it is outside our original scope), we did test different forms of normalization during the learning phase where a contrast between two and three options is possible as in Daviet and Webb. In these simulations the Q-values are learned in an absolute (or unbiased) scale but they are normalized before being fed to the softmax decision function (NB: previous studies we called a similar model the “POLICY model” because the normalization occurs at the decision, not the update; see Bavard et al. 2018; 2021).

**Author response image 6. sa2fig6:** Participants’ accuracy (colored rectangles) and model simulations (black dots) in each condition of the learning phase.

First, we note that all models predict (to some extent) lower accuracy in the 3 versus 2 options decision contexts: the effect is inherent to the softmax decision rule (and indicate that this effect, in absence of the appropriate controls, is only weakly diagnostic of DN). Second, and more relevant for our discussion here, while no model is perfect, it looks like that the DN model produces the simulations that diverge the most from the actual data. This indicates that participants’ behavior (during the learning phase) was not significantly affected by the number of options, beyond what is “mechanically” predicted by the softmax rule. Therefore, it seems that, even in this phase, their behavior was not consistent with a DN process (contrary to what was observed by Daviet and Webb with similar manipulation). These analyses were not planned (and we know in advance that these models are not fully plausible: they could not account for transfer phase preferences!), yet, especially since *eLife*’s rebuttal letters are published, we are happy to leave this information here for future readers questioning the effect of the number of options during the learning phase in our task.

3) In Discussion:a) lines 361-371: This paragraph argues that "We find no evidence for such a [choice set size] effect". As noted in my previous report (and again above), this study is not examining choice set size effects. The size of the choice set in the transfer phase is always binary. Instead, the study is examining the relative coding of learning. This paragraph needs to be re-written so that this is clearly stated. Referring to the results from the transfer task as a "trinary decision context" is confusing.

We rephrased this paragraph to make the description-experience distinction (i.e., the different domain of application of the two models) even clearer:

“Our demonstration relied on the straightforward and well accepted idea that virtually any instantiation of the divisive normalization model would predict a strong choice (description-based task) or outcome (reinforcement-based task) set size effect: all things being equal, the subjective value of an option or an outcome in a trinary decision context should be lower compared that of a similar value belonging to a binary context. We find no evidence for such an effect. In fact, if anything, the subjective values of options belonging to trinary decision contexts were numerically higher compared to that of the binary decision contexts.”

b) "However, it is worth noting that evidence concerning previous reports of divisive normalization in humans has been recently challenged and alternatively accounts, such as range normalization, have not been properly tested in these datasets 29,65."The test for Range normalization in the original Louie et al. dataset is in Webb, Louie andGlimcher 2020, Mng Sci and is rejected in favour of DN. It is not clear what the authors mean by "properly."

Good point, we replace “properly” by “systematically” and rephrased the sentence:

“However, it is worth noting that evidence concerning previous reports of divisive normalization in humans have been recently challenged and alternative accounts, such as range normalization, have not been systematically tested in these datasets^30,65^, although see ^19^ for an exception. »

4) R2.3b In the ex-ante simulations, we randomly sampled both of these parameters from Β(1.1,1.1) distributions, which inherently leads to different predictions in the transfer phase for the binary and trinary contexts, because the choice rate impacts the update of each option. The fact that small differences derive from allowing different learning rates for obtained and forgone outcomes, is confirmed by simulations. We nonetheless kept the original simulations, because they correspond better to the models that we simulated (with two learning rates). The authors are arguing that there is a systematic difference in the predictions for Range WT_86 WB 86 when using two learning rates. This is not an obvious result, but the increase to 500 simulations seems to confirm that it is systematic. This should be clarified in the text. Otherwise, it will be confusing to readers why the predictions change.

We have a different perspective than the Reviewer on this point. The difference is barely noticeable (0.035 relative increase) and we are afraid that adding a lengthily and technical explanation of this effect at this stage of the paper in this journal might distract the reader from the main points and results rather than be useful. Nonetheless, the nerdier readers will have a chance to refer to the rebuttal letters (as well as all our codes and data, openly accessible online) for clarification.

**Author response image 7. sa2fig7:** For the sake of completeness we show again here the two-learning rates-induced discrepancy between the preferences in N86/N50 and W86/W50.

5) From Round 1: line 415 – what is meant by "computational advantage" here? The rational inattention literature, which provides the normative foundation of efficient encoding, places emphasis on the frequency of stimuli (which is completely ignored by range normalization) (e.g. Heng, Woodford, Polania, eLife).Reply: "R2.18 Range and Divisive normalization both provide, to some extent, the computational advantage of being a form of adaptive coding. If the topic interests the Reviewer, we could point to a recent study where their normative status is discussed: https://papers.ssrn.com/sol3/papers.cfm?abstract_id=4320894"The normative argument for relative coding (including both DN and RN) that the authors point to is specific to a learning context (via the exploration/exploitation tradeoff). And even then, it is not an argument that an efficient code should use only the range. Thus it is important to keep in mind that the results reported in the current manuscript may be somewhat specific to the learning setting implemented in the experimental design. Multiple theoretical and empirical papers from across neuroscience and economics have examined the form of normative computation in general settings (Heng et al., cited above; Steverson et al. cited in previous report; Netzer 2009 AER; Robson and Whitehead, Schaffner et al., 2023, NHB) and all have shown that the efficient code adapts to the full distribution of stimuli. It is reasonable to imagine that some physical systems might approximate this with a piecewise linear algorithm based on the range, but this has not been observed in the nervous system (V1, A1, LIP etc.). Of course this can't rule out the possibility that V1, A1 and LIP get it right with a standard cortical computation, but for some reason other areas have to resort to a cruder approximation. But that isn't the obvious hypothesis. Tracking the minimum and maximum – including identifying which is which – seems both unnecessary and complicated in real-world scenarios. Whether (and how much) the results reported in the paper generalize outside this experimental design is an open question.

Across two rounds of reviewing, the Reviewer has spent a lot of justified effort to push us to better clarify that we are working in reinforcement learning framework. As a consequence, we believe it should seem rather appropriate that, when referring to normative advantage, we focus on applications in reinforcement learning.

Concerning the very last point:

“Tracking the minimum and maximum – including identifying which is which – seems both unnecessary and complicated in real-world scenarios. Whether (and how much) the results reported in the paper generalize outside this experimental design is an open question.”

While our position differs from the first part (our intuition is that there is nothing easier and more natural than tracking the max and the min of any contextual variable), we do agree with the second part (generalization in the field is an open question).